# Development and validation of AI/ML derived splice-switching oligonucleotides

Alyssa D Fronk[1,2], Miguel A Manzanares[1,2], Paulina Zheng[1], Adam Geier[1], Kendall Anderson[1], Shaleigh Stanton [1], Hasan Zumrut[1], Sakshi Gera[1], Robin Munch[1], Vanessa Frederick[1], Priyanka Dhingra[1], Gayatri Arun [1] & Martin Akerman [1✉]

## Abstract

Splice-switching oligonucleotides (SSOs) are antisense compounds that act directly on pre-mRNA to modulate alternative splicing (AS). This study demonstrates the value that artificial intelligence/machine learning (AI/ML) provides for the identification of functional, verifiable, and therapeutic SSOs. We trained XGboost tree models using splicing factor (SF) pre-mRNA binding profiles and spliceosome assembly information to identify modulatory SSO binding sites on pre-mRNA. Using Shapley and out-of-bag analyses we also predicted the identity of specific SFs whose binding to pre-mRNA is blocked by SSOs. This step adds considerable transparency to AI/ML-driven drug discovery and informs biological insights useful in further validation steps. We applied this approach to previously established functional SSOs to retrospectively identify the SFs likely to regulate those events. We then took a prospective validation approach using a novel target in triple negative breast cancer (TNBC), *NEDD4L* exon 13 (*NEDD4Le13*). Targeting *NEDD4Le13* with an AI/ML-designed SSO decreased the proliferative and migratory behavior of TNBC cells via downregulation of the TGFβ pathway. Overall, this study illustrates the ability of AI/ML to extract actionable insights from RNA-seq data.

**Keywords** Alternative Splicing; Machine Learning; Splice-Switching Oligonucleotides; Splicing Factors; Triple Negative Breast Cancer
**Subject Categories** Cancer; Computational Biology; RNA Biology

## Introduction

RNA splicing is the mechanism by which introns are removed from newly transcribed pre-mRNA to produce mature mRNA. Alternative splicing (AS), a process by which exonic sequences are differentially skipped or included in the mRNA, allows multiple protein isoforms to be encoded by a single gene. AS regulates many biological processes, such as cell differentiation, cell state reprogramming, and stress response (Ule and Blencowe, 2019). AS also plays a role in cancer, where tumor-specific AS events can drive tumor progression, metastatic transition, and drug resistance, among other hallmarks of cancer (Urbanski et al, 2018). The growing body of evidence reinforcing the importance of AS in cancer highlights an opportunity to target and drug AS events. While AS can drastically change the structure and function of the resulting protein, it is often difficult to identify small molecule drugs that specifically target these protein isoforms. Conversely, modulation of RNA splicing represents a powerful tool for targeting "undruggable" targets, as these treatment modalities can act directly on mRNA or pre-mRNA (Havens and Hastings, 2016).

Splice-switching oligonucleotides (SSOs) are effective in blocking the interaction between splicing factors (SFs) and their pre-mRNA targets in the nucleus to change AS outcomes (Havens and Hastings, 2016). SSOs can target mis-splicing or disease-specific AS events ahead of translation, allowing the SSO to potentially modulate downstream protein activity without the need to directly inhibit protein function (Havens and Hastings, 2016). Recent advances in antisense chemistry and delivery have led to successful clinical translation of oligonucleotide drugs in muscle and CNS diseases (Syed, 2016; Finkel et al, 2017; Han et al, 2020; Kim et al, 2019; Centa et al, 2020; Wagner et al, 2021). While SSOs for monogenic CNS diseases have shown success in clinical trials, most antisense oligonucleotides currently under investigation are for the treatment of cancer (Xiong et al, 2021), including but not limited to SSOs targeting AR-V7, PKM, and BCL-X pre-mRNAs (Yamamoto et al, 2015; Ma et al, 2021; Li et al, 2016). Typically, SSOs are developed through the costly and time-consuming process of microwalks which require 100- 200 tiling oligos to be tested manually for activity. To unlock the innovative potential of SSO treatments, there is a need for efficient, scalable technologies to discover novel drug targets and develop SSO compounds for the treatment of cancer and other diseases in which AS misregulation plays a key role (Dvinge et al, 2016; Kahles et al, 2018; Park et al, 2019; Urbanski et al, 2018).

In this study, we utilized and developed artificial intelligence/machine learning (AI/ML) algorithms for the discovery of novel therapeutic SSOs based on the analysis of RNA-seq data. We trained XGboost (Zhao et al, 2022; Sheridan et al, 2016) trees using sequence-specific binding profiles of SFs, along with spliceosome assembly information based on SF-RNA and SF–SF interactions. We also used Shapley (SHAP) and out-of-bag (OOB) analyses to prioritize the most predictive SFs. As a result, our model not only

[1]Envisagenics, Inc., Long Island City, NY 11101, USA. [2]These authors contributed equally: Alyssa D Fronk, Miguel A Manzanares. ✉E-mail: makerman@envisagenics.com

predicts the optimal binding position of SSOs that modulate AS, but also informs the identity of SF regulatory networks under steric inhibition on a given RNA, allowing for more transparent and actionable SSO predictions.

The primary goal of this study was to leverage an AI/ML approach to efficiently identify and develop functional SSO compounds to modulate disease-relevant AS events. Once identified, our secondary goal was to validate the SSOs both retrospectively and prospectively. We highlight a new SSO compound identified to modulate a novel AS event in TNBC models. We present experimental evidence showing that a novel target, *NEDD4L* exon 13 (*NEDD4Le13*), was successfully predicted to be disease-relevant using AI/ML, and that AI/ML-designed *NEDD4Le13* SSOs can promote cell death specifically in TNBC by modulating the TGFβ pathway. Overall, these data lend credence to the use of AI/ML for drug discovery by identifying novel and verifiable drug targets in silico and provide evidence for one such therapeutic candidate for TNBC.

## Results

### An AI/ML model to predict SSO binding sites for AS modulation

To identify productive SSO binding sites for AS modulation, we used splicing regulatory information to train various types of AI/ML algorithms. The resulting models were able to estimate the probability that delivering SSOs to a specific RNA-binding position would elicit AS of its most proximal exon (Fig. 1).

We combined three sources of splicing regulatory information in the model's training features: the positions of potential SSO binding sites relative to the alternative exon, the identity and binding motif scores of SFs potentially blocked by SSOs (Lambert et al, 2014; Paz et al, 2014; Ray et al, 2013), and the interaction of such SFs with other SFs within the spliceosome (Akerman et al, 2015) (see Methods for details). Upon consolidation of these three data sources, we combined SFs into 83 non-exclusive SF clusters (SFCs), based on spliceosomal functional annotations and SF–SF binding probabilities (Akerman et al, 2015) (Fig. 1; Dataset EV1). Because the spliceosome is a highly dynamic complex, SFs can interact with different partners depending on the stage of the splicing process. It is crucial to take this plasticity into account when assigning protein–protein interactions (PPIs). By using probabilistic PPIs within the spliceosome, it is possible to account for the regulatory effect of the SF and its role in the splicing cycle (Akerman et al, 2015). Therefore, each SFC was composed of physically interacting SFs attributed to a given splicing-related function such as membership to spliceosomal subcomplexes (e.g., U1 snRNP), RNA-binding specificity (e.g., AG binding), regulatory outcome (e.g., repressor), and others (Dataset EV1). Since SFs perform multiple functions in splicing regulation, they were non-exclusive and permitted to appear in more than one SFC. The advantage of using SFCs as predictive features in lieu of SF binding scores alone is that they reduced matrix sparsity and zero-inflation, while capturing functional aspects of the spliceosome. By grouping SFs into SFCs, we provide a more intuitive context for downstream biological explainability (see Methods for details).

Training labels were derived from a massively parallel splicing minigene reporter assay called MFASS (Cheung et al, 2019), which quantifies the effect of single nucleotide variants (SNVs) on AS outcomes, therefore mimicking the impact of SF-blocking by SSOs. The MFASS dataset assessed the effect of 27,733 SNVs extracted from the ExAC database on the splicing of 2198 distinct human exons. Of these, 14,130 SNVs occurred in exons, 7938 in the intronic region upstream of the 3′ splice site, and 6271 in the intronic region downstream of the 5′ splice site. These three RNA regions present striking differences in sequence composition and have evolved to play distinct roles in splicing regulation (De Conti et al, 2013). Therefore, we developed three independent AI/ML models to account for the unique regulatory properties of the exonic and upstream/downstream intronic elements (Fig. 1). The final set of training labels was composed of 2005 "positive" SNVs inducing significant AS changes with a "delta percent splice in" of dPSI ≥ |0.25|, and a pool of 14,936 "negatives" intronic SNVs with no effect on AS (dPSI ≤ |0.05|) (Dataset EV2).

To appropriately select a training method, we used this training set to train six different models, including Logistic Regression, SVM, Random Forest, Gradient Boost, AdaBoost, and XGboost (see Methods for details). While all the models used the exact same set of training features (Dataset EV2), in all cases (i.e., -exon, -upstream, and -downstream, Fig. 1), the best-performing model type was XGboost trees; therefore, it was selected for implementation (Appendix Table S1). Interestingly, XGboost-downstream was the top-performing model with an AUC of 0.95, followed by XGboost-upstream with an AUC of 0.88, and finally, XGboost-exon with an AUC of 0.60. The sensitivity and specificity were high for both XGboost-downstream (0.92 and 0.93, respectively) and XGboost-upstream (0.84 and 0.75), although for XGboost-exon only the specificity was relatively high (0.71) while the sensitivity was poor (0.40) (Fig. EV1A, see Methods for details). Notably, both intronic XGboost classifiers clearly outperformed the exonic one, suggesting that it is easier to predict productive SSO binding sites in introns vs exons. This could be due to the fact that, unlike exons, introns are not subjected to protein-coding constraints, thus, regulatory information may be easier to identify in introns using AI/ML (Wang and Burge, 2008). In practice, intronic regions near the splice sites are preferable for SSO targeting compared to exons, since introns are only present in nuclear pre-mRNA, while exons are both present in pre- and mRNA, potentially increasing the chances of off-target effects or translational perturbation. For example, the SSO Nusinersen that treats Spinal Muscular Atrophy targets the intron downstream of exon 7 in the *SMN2* pre-mRNA, to block binding of the splicing repressor hnRNPA1 (Singh and Singh, 2018).

After training an XGboost model with the MFASS dataset, we tested it with an equivalent but completely independent dataset called Vex-seq (Adamson et al, 2018). Vex-seq consists of 1226 qualifying variants that were experimentally identified to have an impact on pre-mRNA splicing using a high-throughput reporter assay. As a result, The XGboost-downstream model classified Vex-seq data with an AUC of 0.98, XGboost-upstream with an AUC of 0.86, and XGboost-exon with an AUC of 0.66 (Fig. EV1B). Altogether, these data support the usefulness of both intronic XGboost models to predict SF-binding perturbation positions critical for AS regulation and useful for SSO targeting.

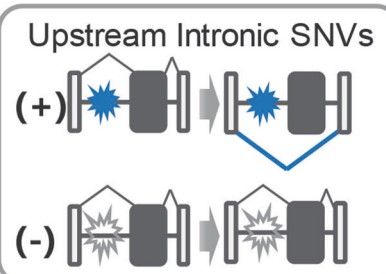
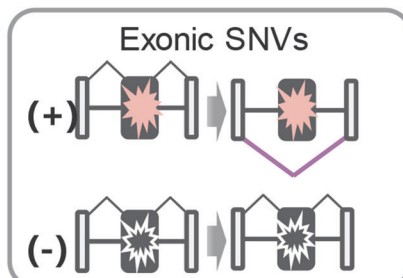
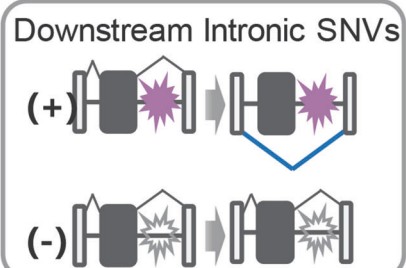

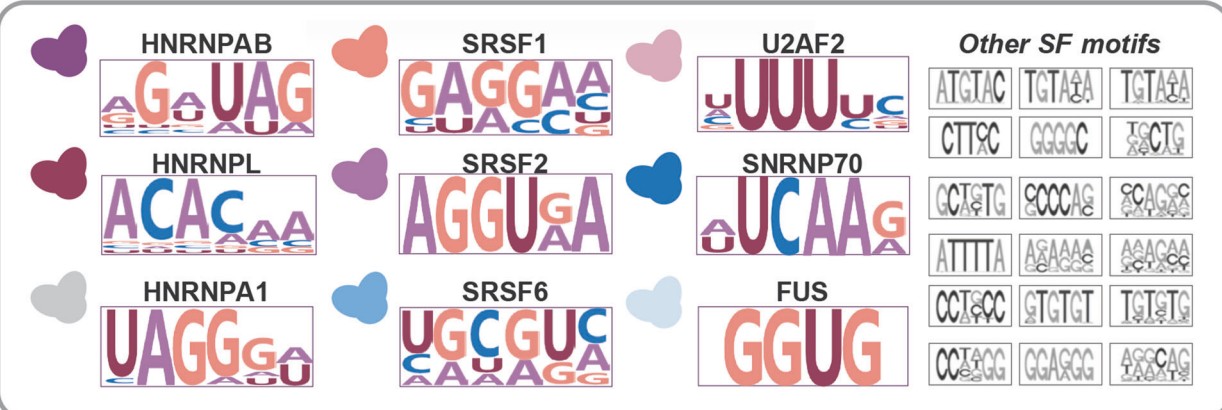

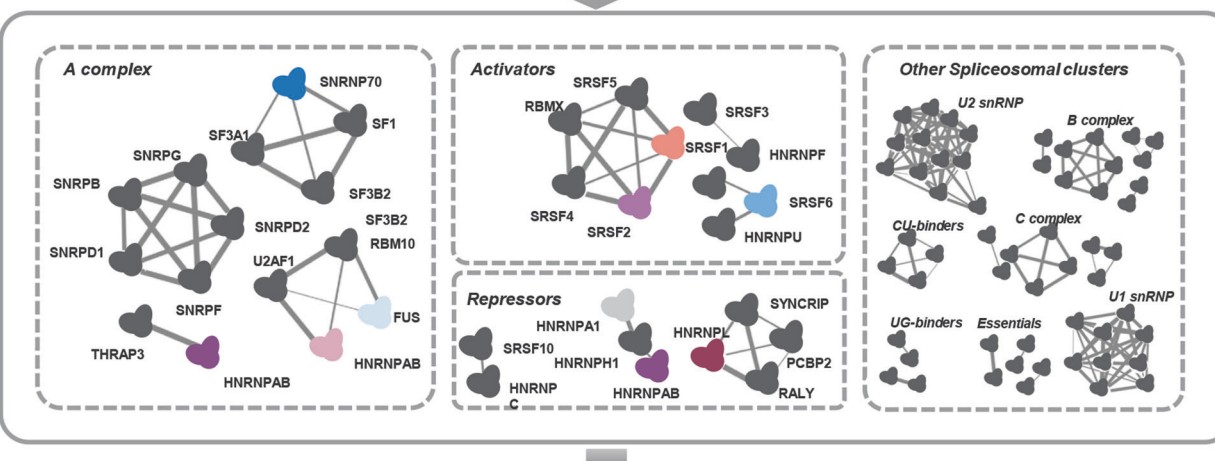

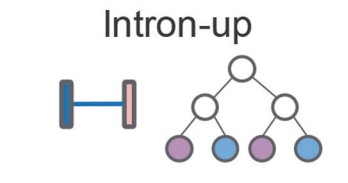
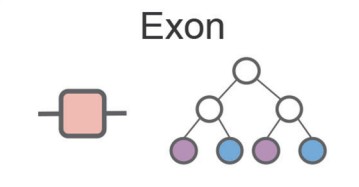
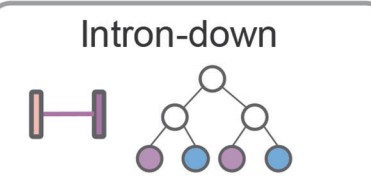

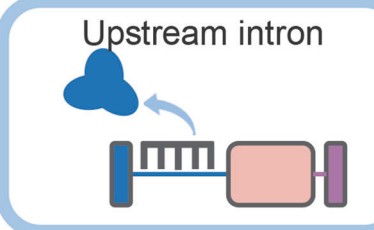
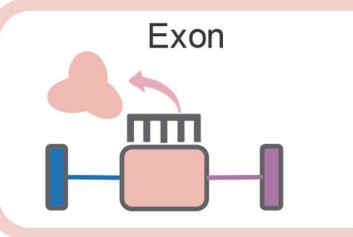
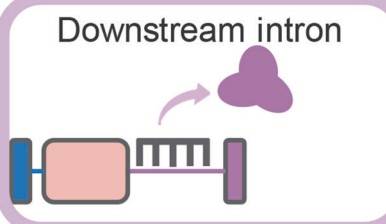

**Figure 1. Overview of AI/ML model for SSO drug target prediction.**

XGboost trees were trained with splicing regulatory information including SF-RNA binding profiles, SF–SF interactions and position-specific splice-switching SNVs. Predictive features were extracted in a context-specific manner to generate three independent XGboost models for upstream introns, exons and downstream introns.

## Exonic and intronic predictive models require different feature combinations

Exons and introns present striking differences in function, evolution, and modes in which they interact with the spliceosome (De Conti et al, 2013). To test the independent contribution of SFCs to the XGboost-exon, -up and -down models, we tested the ability of each SFC to differentiate between "positives" and "negatives" using the Wilcoxon test with Holm–Sidak $p$ value adjustment (Guo and Romano, 2007). We observed that the three sequence types were characterized by different subsets of significant SFCs (Dataset EV1). For instance, the "distance to splice site" SFCs were highly significant in introns (adj.$p$ val $<3.4 \times \times 10^{-7}$) but not exons (adj.$p$ val $\leq 0.521$). This observation was expected and agrees with many studies showing intronic sequences around the splice sites to be enriched with SF binding sites important for AS regulation (Wang and Burge, 2008; Yeo et al, 2007; van Nostrand et al, 2016; Yee et al, 2019). In addition, serine and arginine-rich (SR) proteins and activators were highly significant SFCs in exons (adj.$p$ val $\leq 2.14 \times 10^{-8}$ and adj.$p$ val $\leq 4.91 \times 10^{-5}$, respectively) but not introns (0.003 $\leq$adj.$p$ val $\leq 0.98$), consistent with several studies describing the role of SR proteins as splicing activators that bind exonic splicing enhancers (Lin and Fu, 2007; Jeong, 2017; Wang et al, 2005). In upstream introns, SF binding sites such as U-rich (adj.$p$ val $\leq 6.64 \times 10^{-7}$) and CG-rich (adj.$p$ val $\leq 2.1 \times 10^{-6}$) motifs were highly significant. Interestingly, CG-rich introns have been associated with weak 3' splice sites and polypyrimidine tracts, which are important for the regulation of alternative exons (Murray et al, 2008; Wagner and Garcia-Blanco, 2001). These polypyrimidine tracts, which are intrinsically uridine-rich, are known for attracting several SFs (Barreau et al, 2006), including members of the A complex such as U2AF2, known for its crucial role in 3' splice site recognition and exon inclusion (Graveley et al, 2001; Warnasooriya et al, 2020; Singh et al, 2000). Accordingly, the "A complex" SFC was more significant in upstream (adj.$p$ val $\leq 0.0006$) and downstream introns (adj.$p$ val $\leq 3.62 \times 10^{-13}$) than exons (adj.$p$ val $\leq 0.756$). Other highly significant SFCs in downstream introns were "repressors" (adj.$p$ val $\approx 0$) and members of the hnRNP family (adj.$p$ val $\approx 0$) known to interact with intronic splicing silencers to inhibit exon inclusion (Geuens et al, 2016). In addition, SFCs corresponding to hnRNP binding motifs were highly significant, including UG-rich (adj.$p$ val $\approx 0$), CU-rich (adj.$p$ val $\leq 1.80 \times 10^{-14}$), and CA-rich (adj.$p$ val $\leq 4.11 \times 10^{-13}$) elements. The U1 snRNP SFC was also highly significant in the downstream intron (adj.$p$ val $\leq 1.84 \times 10^{-10}$). In summary, a total of 2, 5, and 14 SFCs showed adj.$p$ val $\leq 1.0 \times 10^{-6}$ in exons, upstream and downstream introns, respectively. These results suggest that the training data used by our AI/ML models appropriately captures regulatory information and recapitulates many observations made by researchers in the past.

## Feature selection methods add explainability to prioritize key SFs blocked by SSOs

To properly develop effective SSOs it is necessary to reveal the identity of SFs to be displaced by these SSOs. Knowledge of these specific SFs is not only crucial for biological explainability, but also to facilitate experimental drug design, understand the mechanism of SSO action, and integrate discoveries with other data types (e.g., CLIP-seq). To this aim, we derived a feature selection approach based on two feature prioritization methods, Shapley additive explanation analysis (SHAP) (Lundberg et al, 2020) and out-of-bag data analysis (OOB) (Breiman, 2001). SHAP is a game theory approach that estimates the marginal contribution of predictive features across bootstrapped decision trees. A key advantage of SHAP over other feature importance inference methods is that it is unaffected by the order in which features are randomly chosen by tree models; thus, it is a robust tool for the explainability of the primary information driving predictive efficiency in AI/ML (Lundberg et al, 2020). OOB is a method of measuring feature importance in tree-based models (e.g., XGboost). It uses subsampling with replacement to estimate feature importance as the mean prediction error of models with vs. without a given feature (Breiman, 2001). OOB complements SHAP analysis by performing independent feature prioritization.

We applied SHAP/OOB to weigh the contribution of every SFC to XGboost-upstream and XGboost-downstream, as these two models substantially outperformed the XGboost-exon model. The SHAP/OOB analysis revealed "distance to splice site" information as a major driver of feature predictability, whereby positions closer to the splice sites show greater potential for AS alterations (Appendix Fig. S1A,C). This observation suggests that the position of perturbing factors (i.e., SNVs and SSOs) relative to the splice sites is sufficient to explain much of the AS outcome. Despite its strong predictive power, "distance to splice sites" alone does not provide sufficient explainability regarding the role of specific SFs. To avoid the dominating effect of "distance to splice site" and allow SHAP/OOB to best prioritize other explainable features, we retrained XGboost-upstream and -downstream, this time without any "distance to splice site" information. As a result, we observed that despite a proportional reduction in predictive efficacy, XGboost-downstream still performed well with an AUC of 0.80, while XGboost-upstream showed borderline predictive power with an AUC of 0.67 (Fig. EV1C). Additionally, this highlights additional SFCs that drive predictability, such as SFCs corresponding to hnRNP binding motifs in downstream introns and members of the spliceosomal A complex in upstream introns (Appendix Fig. S1B,D).

Next, to prioritize specific SFs most likely to bind RNA and regulate AS at every nucleotide position, we used a distributed min-max strategy that combined SHAP/OOB values with SF binding motif scores (Lian et al, 2022). To start, we selected the most predictive SFs in each SFC as the top 25% ranked SHAP/OOB values combined with the best binding motif scores at percentile score >50, only at positions where XGboost predicted an AS effect probability of at least 0.5. To test the usefulness of this SF selection method and evaluate the correspondence between measured explainability and evidence of binding in vivo, we compared our top predicted SFs to external eCLIP data from the

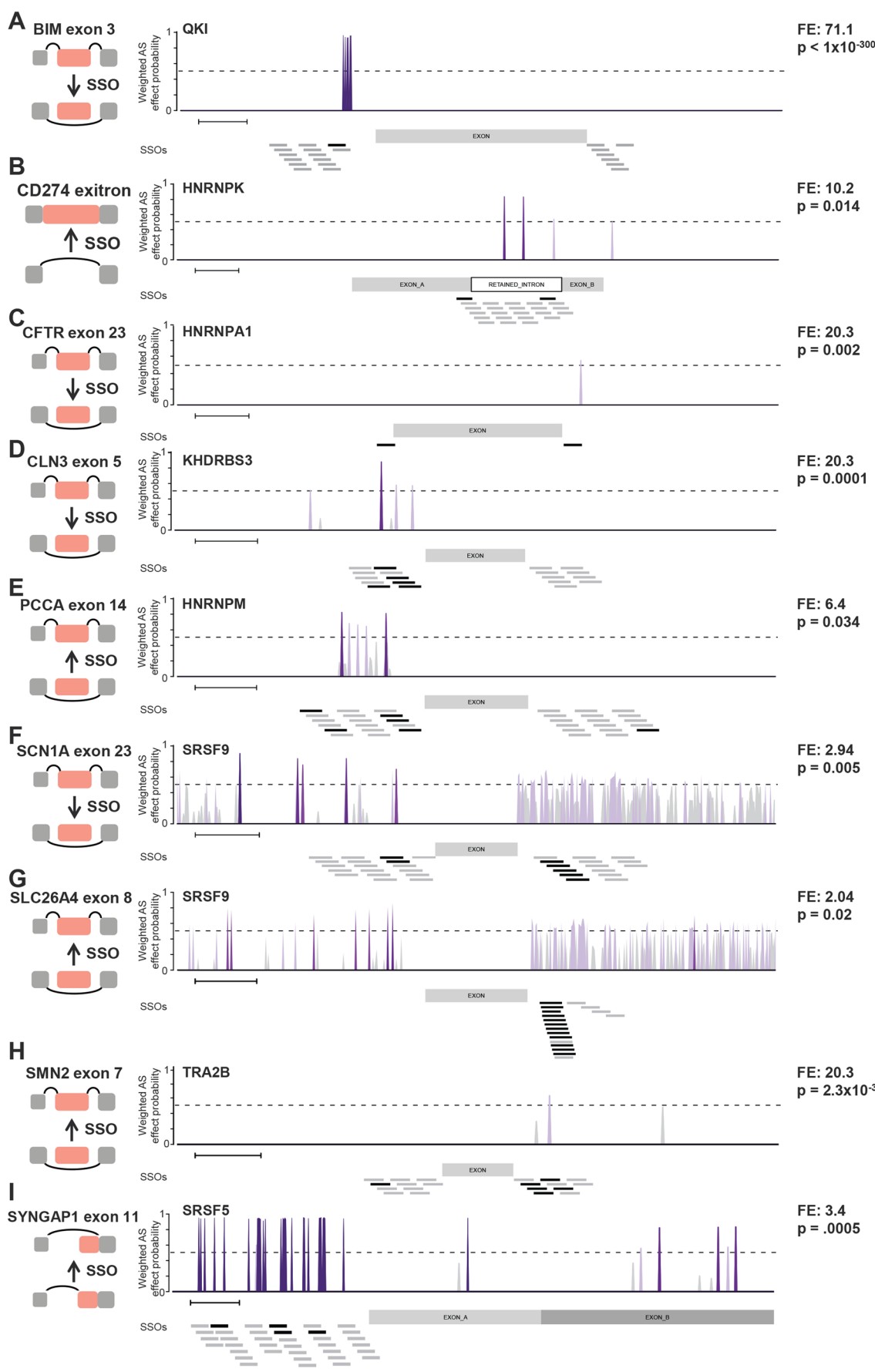

**Figure 2. Retrospective validation and identification of regulatory SFs in known SSO modulated AS events.**

(A–I). Left: diagram of splicing event analyzed, including the direction of splice-switching induced by the SSO. Middle: SF-specific AS effect probability scores as determined by the XGboost model. Black lines indicate the position of previously established functional SSOs. Gray lines indicate the position of previously published ineffective SSOs. Scale bars: 50 nucleotides. Right: Fold enrichment and p-value calculated using a hypergeometric test for the top SF predicted to be regulated by the functional SSOs.

ENCODE project (van Nostrand et al, 2020). After performing predictions on 2124 introns with eCLIP coverage, we identified a significant overlap between specific SFs predicted by the model to influence AS, and enrichment for eCLIP peaks (Appendix Fig. S1E,F). Using the odds ratio as a metric, we then proceeded to test a wide range of cutoffs for each variable to determine the set of conditions that best enriches predicted hits for eCLIP peaks for each individual SF. Generally, we found that using a highly stringent cutoff for the splicing effect probability allowed for the most robust enrichment with the eCLIP data (Appendix Fig. S1G). Overall, we have shown that AI/ML can provide biological explainability, which is crucial for the development of novel therapeutics. Specifically, we have seen that predictions made by the XGboost model can be traced back to one or multiple SFs, and that these predicted SF binding sites are enriched for eCLIP peaks in independent datasets, confirming the binding of those SFs at those locations.

## XGboost model retrospectively identifies SSO known to modulate AS events

Several SSOs have been previously identified to regulate specific AS events, typically through labor-intensive SSO microwalks, where SSOs are tiled along an event and tested in vitro, using RT-PCR or qPCR as a readout. We performed a literature search for events with established SSOs. Specifically, we looked for functional intronic SSOs (as our models for upstream and downstream introns perform particularly well), and for studies that performed true SSO walks. We excluded SSOs that were fully within the exon or overlapping the splice site. We found and analyzed nine such events where SSOs were identified and confirmed to modulate the splicing of the target exon (Liu et al, 2017; Feng et al, 2022; Lim et al, 2020; Han et al, 2020; Hua et al, 2008; Kim et al, 2022; Centa et al, 2020). We applied our XGboost model to these events to retrospectively validate the model as well as identify the SFs likely to regulate the SSO function. In brief, we applied the XGboost model to the upstream and downstream introns surrounding the alternative exon in each event, identifying the AS effect probability and the most predictive SFs at each position using our SHAP/OOB approach (Dataset EV3). We then performed an in silico SSO walk to analyze every potential SSO in an event and used a hypergeometric test to measure the enrichment of "hits" (nucleotides with high AS effect probability) within an SSO and the SFs predicted to regulate those hits. We ranked the SSOs by their SF enrichment scores to find the SFs and SSOs most likely to be functional. We also evaluated the performance of the model on each individual splicing event, using the enrichment scores as a metric of functional SSO prediction. For example, exon 3 inclusion in *BIM* has been shown to promote resistance to tyrosine kinase inhibitors (TKIs) in certain cancers, but SSOs that promote exon 3 skipping re-sensitize chronic myeloid leukemia cells to TKIs.

Several such SSOs in the upstream intron of *BIM* exon 3 (*BIMe3*) have been identified (Liu et al, 2017). We applied our XGboost model to the *BIMe3* event and, using the method described above, found that the top two hits in our analysis are included in the functional SSOs that were previously described (Liu et al, 2017). Furthermore, we found that nucleotides within this functional SSO were significantly enriched for hits predicted to be regulated by the SF QKI (fold enrichment = 71.1, $p$ val $<1 \times 10^{-300}$, Fig. 2A). Overall, we found that our model performed extremely well on this splicing event, with an AUC of 0.95 (Fig. EV2). We analyzed the eight additional published SSOs and found that our XGboost model predicts the functionality of the published SSOs and finds the SFs that are significantly enriched for hits within the functional SSOs (Fig. 2B–I). We evaluated the performance of the eight events where both positive and negative SSOs were tested and found that four had strong predictive power (AUC >0.7), two had borderline predictive power (AUC >0.6), and two had weak predictive power (AUC >0.5) (Fig. EV2). This variability is likely due to the relatively small number of SSOs tested for each event. One of the advantages of using our in silico model is the ability to test and evaluate all possible SSOs in a splicing event, and incorporating the SF enrichment scores allowed us to achieve clear, high predictive power across several events. Ultimately, this validates our XGboost model and feature selection pipeline, highlighting its ability to not only find functional SSOs but also predict the SFs that are likely blocked by these SSOs.

## Identification of novel SSO drug targets candidates in RNA-seq data from TNBC patients

To take a prospective approach and demonstrate the unique potential of AI/ML in identifying new drug targets, we sought to investigate the cancer genome atlas (TCGA), one of the most popular sources of RNA-seq data in cancer studies (Koboldt et al, 2012). Our premise was to accurately identify a novel SSO drug target in a highly accessed dataset like TCGA, to prove that AI/ML can discover new targets and extract value from public data, even if it has been used thousands of times in the past.

TNBC is one of the most aggressive forms of breast cancer (Garrido-Castro et al, 2019). TNBC has been shown to be transcriptionally distinct when compared to other subtypes of breast cancer, particularly with respect to luminal breast cancer, which overexpresses hormone receptors (Kahles et al, 2018). Interestingly, we observed that 81.5% of TNBC samples in TCGA presented copy-number or transcriptional alterations in at least one regulatory SF, illustrating the extent of variability and potential damage to the spliceosome in TNBC (Appendix Fig. S2). It has been shown that TNBC progression and survival depends on SFs like SRSF1 and TRA2B (Leclair et al, 2020; Anczuków et al, 2015; Du et al, 2021). Based on the unmet need and the strong scientific premise, we investigated novel AS events critical for TNBC

progression and developed AS-correcting SSOs for these potential targets (Kahles et al 2018).

We first quantified AS from RNA-seq data using our target selection platform called SpliceCore® (Appendix Supplementary Methods; Appendix Figs. S3, S4). In brief, SpliceCore uses two previously established algorithms to align RNA-seq data and quantify the PSI of each AS event (SpliceTrap, (Wu et al, 2011)), and estimate differential splicing changes (dPSI) between case and control datasets (SpliceDuo, (Anczuków et al, 2015)). We applied this pipeline to TCGA breast cancer data and identified 8725 AS events expressed in TNBC basal tumors ($n = 148$) but not normal breast tissue ($n = 108$), and 8912 AS events unique to TNBC basal tumors when compared to luminal tumors ($n = 666$) (Fig. 3A; Dataset EV4). To independently confirm basal-specific AS changes identified in TCGA, we generated RNA-seq data in triplicate for two representative basal cell lines that emulate TNBC (HS578T and BT549) and two representative luminal cell lines as a control (MCF7 and T47D). Cross-comparison of AS profiles in basal vs. luminal cell lines resulted in the confirmation of 250 AS changes originally found in TCGA (Fig. 3B; Dataset EV4). Because these events were consistent across multiple data sources, it is more likely that they are playing an important role in the disease. These candidates were further prioritized with additional attributes available in SpliceCore, including the extent of AS (measured as the dPSI), the prevalence in TCGA samples, and biological significance of the underlying genes in cancer pathways. While some of the top candidates have been previously reported to play a role in cancer (*FLNB, MAP3K7, NFYA,* and *ESYT2* (Li et al, 2018, 2021; Dolfini et al, 2019; de Miguel et al, 2016)) others were identified to be breast cancer-relevant for the first time (*NEDD4L, MARK2, ABI1*). The combination of these parameters resulted in a short-list of 7 candidates with potential for further investigation as SSO targets, and we confirmed the AS changes using RT-PCR in a panel of normal and breast cancer cell lines (Luminal cell lines: MCF7 and T47D; TNBC cell lines: MDA-MB-468, MDA-MB-231, BT549, and HS578T) (Fig. 3C).

## NEDD4Le13 was identified as a potential candidate for therapeutic SSOs in TNBC

AS of *NEDD4Le13* was selected by SpliceCore as a top candidate, showing prominent exon skipping in 109/169 (64%) of TNBC patients, as well as at the RNA and protein levels in breast cancer basal cell lines (Figs. 3C, 4A). NEDD4L belongs to the ubiquitin ligase family of proteins that play a role in the mono-ubiquitination of several important proteins involved in cellular homeostasis and signaling, including proteins in the TGFβ pathway. At the protein level, skipping of NEDD4Le13 was predicted to remove a short loop region next to the second WW domain, a region important for protein–protein interaction (Fig. 4B,C) (Gao et al, 2009; Aragón et al, 2012). The loop also contains an accessible threonine residue that is likely phosphory-lated by Protein Kinase A (Snyder et al, 2004). Since this region has been shown to critically interact with SMAD proteins to regulate TGFβ signaling (Gao et al, 2009), we hypothesized that loss of this short loop through AS of exon 13 deregulates TGFβ signaling to promote tumor progression. Accordingly, we observed that breast cancer patients expressing full-length *NEDD4L* had significantly better overall survival compared to patients with predominant

skipping of exon 13 (Fig. 4D). Tumor-type stratification of *NEDD4Le13* in TCGA data showed the PSI of *NEDD4L* to be significantly lower in TNBC tumors compared to normal breast tissue ($p = 2.5 \times 10^{-6}$), luminal tumors ($p < 2.22 \times 10^{-16}$), normal-like tumors ($p = 7.9 \times 10^{-7}$), and HER2+ breast tumors ($p = 4.9 \times 10^{-14}$), indicating that the skipping isoform is more dominant in TNBC tumors (Fig. 4E). We also observed small but significant changes in *NEDD4L* gene expression across the subtypes, particularly between basal and luminal tumors and basal and normal tissue (Fig. 4F). Given this data along with the important role of ubiquitin ligases in signal transduction pathways, and NEDD4L in particular in regulating TGFβ signaling, we prioritized *NEDD4Le13* as a lead target for SSO design.

## AI/ML identified an optimal SSO to promote *NEDD4Le13* inclusion and explained the underlying AS regulatory network

Because the skipping isoform of *NEDD4L* is dominant in TNBC, we set out to design an SSO that would reverse the skipping phenotype and promote exon inclusion. This is non-trivial, because causing exon inclusion requires blocking a splicing repressor, in contrast to promoting exon exclusion, which could theoretically be achieved by simply blocking a splice site. So, we utilized the XGboost model to find the optimal binding sites for SSOs to promote *NEDD4Le13* inclusion, as well as to identify the underlying SF network regulating this AS event. Feature prioritization using SHAP/OOB and SF enrichment analysis, as described previously, revealed that the two SFs most likely to regulate AS of *NEDD4Le13* were HNRNPL and SRSF7 (Fig. 5A).

Using the SF enrichment analysis previously described, we found that the upstream intron of the *NEDD4Le13* event contained the majority of top hits, and HNRNPL was enriched in the majority of SSOs tested in silico in the upstream intron, including all seven SSOs tested in vitro (Fig. 5A, black bars). Importantly, these two SFs showed confirmatory ENCODE eCLIP peaks (van Nostrand et al, 2020) overlapping with predicted hits. Particularly, HNRNPL presented a strong binding signal in four eCLIP replicates, in stark contrast to other well-known SFs like PCBP1, which was not predicted to regulate *NEDD4Le13* splicing (none of the in silico-tested SSOs were enriched for PCBP1) and did not show any confirmatory eCLIP peaks (Fig. 5A). We also investigated protein–protein interactions (PPIs) between HNRNPL and SRSF7 and found the probability of these two SFs directly binding was high ($P_{HNRNPL-SRSF7} = 0.784$). Of note, PPI predicted between HNRNPL and SRSF7 was among the top interactions within the full network of HNRNPL interactions that included 601 proteins (Fig. 5B). Finally, we found that *HNRNPL* and *SRSF7* were both significantly overexpressed in basal tumors when compared to other breast cancer subtypes and to normal tissue samples (Fig. 5C,D). These subtype-specific changes in SF expression highlight their potential to regulate tumor-specific AS events like NEDD4Le13 in a cell-type-specific manner, as TNBC tumors feature a mesenchymal phenotype pathologically compared to the epithelial phenotype of luminal tumors.

To confirm the impact of SRSF7 and HNRNPL on *NEDD4Le13*, we knocked down the expression of both RBPs individually and simultaneously using specific siRNA in MCF10A and MDA-MB231 cell lines. Knockdown efficiency was observed by Western blot

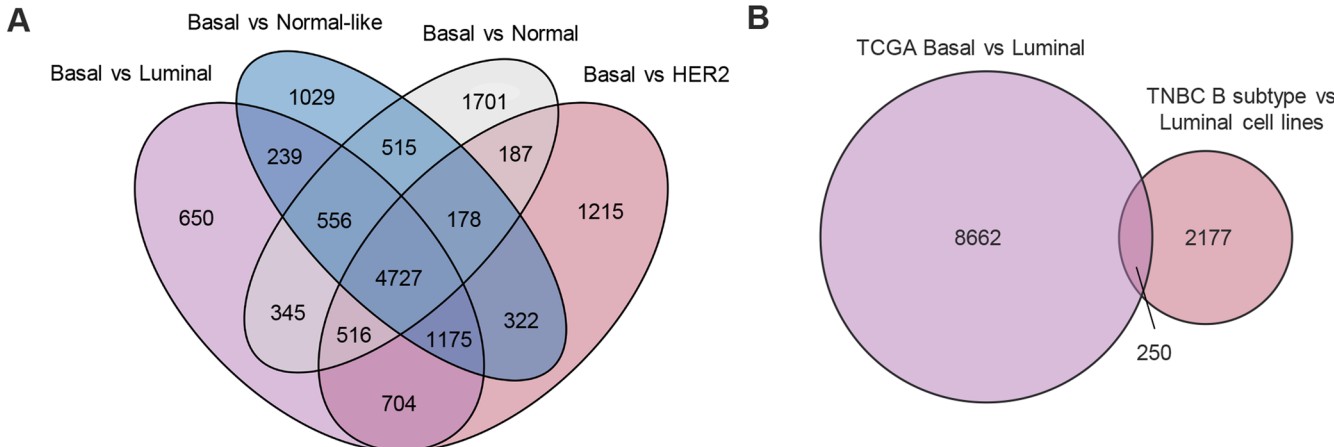

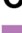

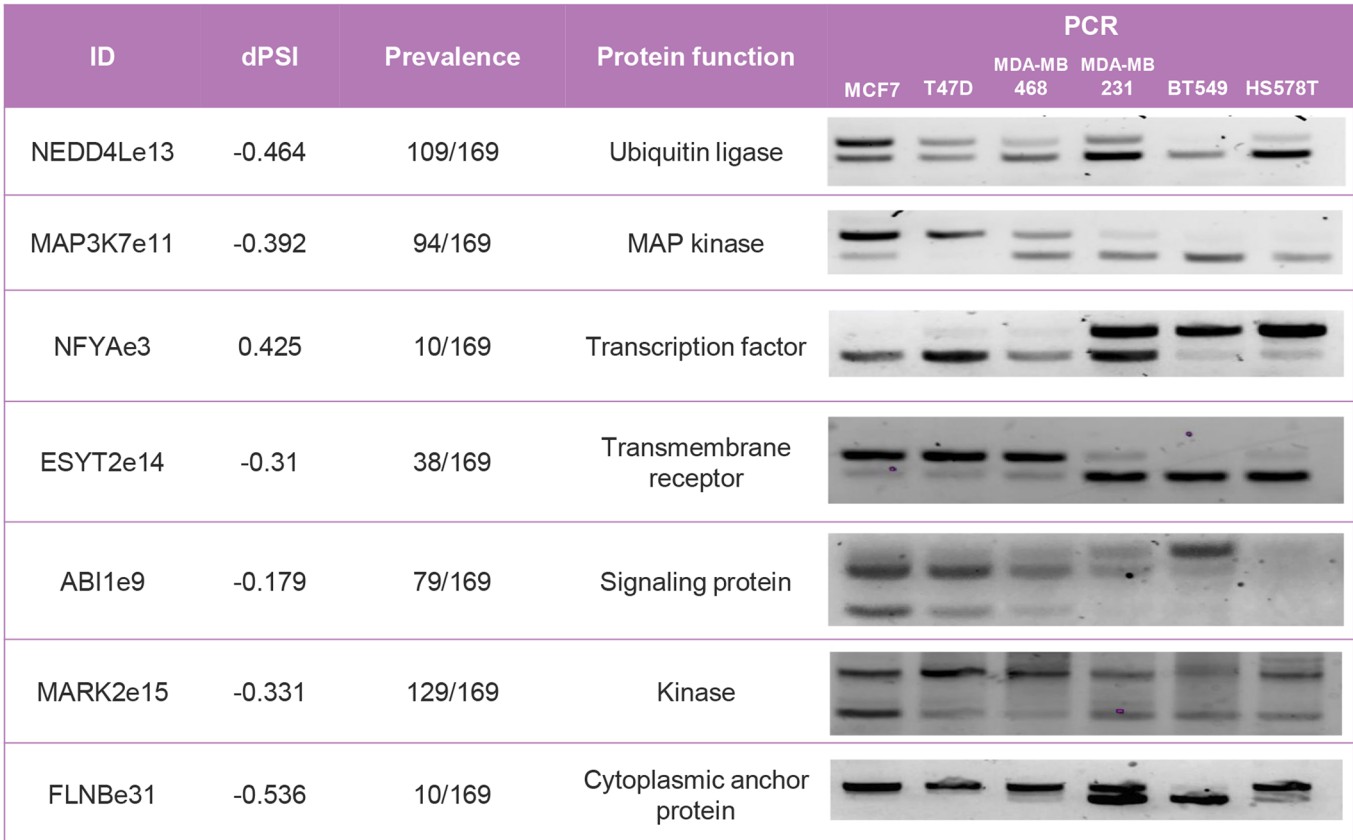

**Figure 3. SpliceCore identified disease-specific, biologically relevant AS events in TNBC.**

(A) Overlap of AS changes across breast cancer tissue types in TCGA. (B) Overlap of AS changes identified in TCGA and cell line RNA-seq data. (C) Top seven SpliceCore targets identified in TNBC. The table shows dPSI values for basal vs. luminal tumor cross-comparisons, prevalence across 169 TNBC samples, a function of the target candidates, and PCRs with AS changes. Source data are available online for this figure.

analysis (Fig. EV3A). We confirmed that when SRSF7 and/or HNRNPL were depleted in MDA-MB-231 cells, NEDD4Le13 inclusion levels significantly increased (Fig. EV3B,C). No significant effects were detected in MCF10A cells. To experimentally validate the predicted SSO binding sites, a list of 20- to 22-mer sequences spanning locations across the highest predictive scores

that overlapped with ENCODE HNRNPL peaks was generated. We also generated a list of seven alternative SSOs in the downstream intron that were not predicted to regulate NEDD4Le13 splicing, to use as negative controls (Fig. 5A; Appendix Table S2). These oligos were chemically modified to enhance stability and nuclease resistance using second-generation antisense chemistry consisting

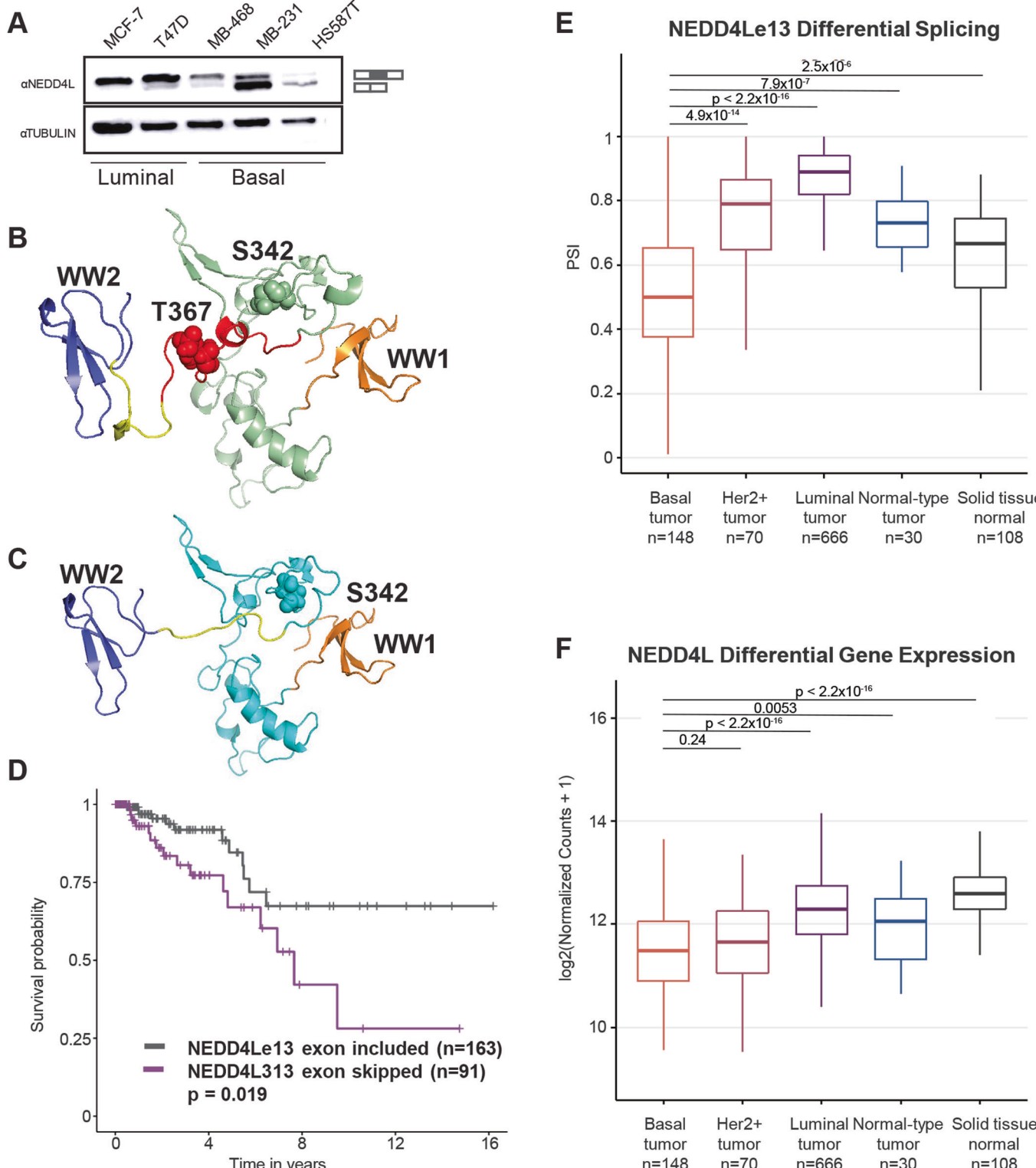

of a phosphorothioate backbone and uniformly modified 2′ methoxyethane (2′MOE) sugar modification. Chemically synthesized and purified oligonucleotides were subjected to functional assays in breast cancer cell lines. Out of the seven sequences predicted to have splice-switching effects, SSO-0205 was found to promote the strongest *NEDD4Le13* inclusion in MDA-MB-231 cells. Treatment with SSO-0205 caused an average of 67% exon inclusion in three independent experiments, compared to 15% inclusion in the lipofectamine control group (Fig. 6A,B). Additionally, MDA-MB-231 cells treated with SSO-0205 showed

◀ **Figure 4. SpliceCore platform identifies NEDD4Le13 as a potential therapeutic target for TNBC.**

(A) Western blot showing the expression of the NEDD4L full-length (top band) and exon 13 skipping (bottom band) isoforms across a panel of breast cancer cell lines. (B) 3D model of amino acids 193–418 in NEDD4L from the transcript, including exon 13. WW domains 1 and 2 are in orange and blue, respectively. Phosphorylation of S342 and T367 are shown as spheres. Red indicates the region encoded by exon 13. The yellow-colored region in both models have a root mean square deviation (RMSD) of 4.271. (C) 3D model of the same region of NEDD4L when exon 13 is skipped. The exclusion of 20 amino acids in the loop connecting the WW1 and WW2 domain alters the helix in the proximity of the WW2 domain with an RMSD of 4.271, whereas no significant RMSD (0.188) was obtained when the two full models were superimposed. (D) Survival curves showing a significant difference in survival between patients where *NEDD4Le13* is included and *NEDD4Le13* is skipped. *NEDD4Le13* exon included $n = 163$, *NEDD4Le13* exon skipped $n = 91$, $p$ value calculated using a log-rank test. (E,F) *NEDD4L* Percent Spliced In (PSI) (E) gene expression data (F) from the TCGA BRCA samples stratified into subtypes. Basal tumor samples $n = 148$, Her2+ tumor samples $n = 70$, Luminal tumor samples $n = 666$, Normal-type tumor samples $n = 30$, Solid tissue normal samples $n = 108$. $P$ values were calculated using a Wilcox test. The box plot center line indicates the median value, upper and lower box edges correspond to the 25th and 75th percentiles, and the whiskers extend to 1.5 * interquartile range from the box edge. Source data are available online for this figure.

substantial *NEDD4Le13* RNA-seq read coverage, compared to the untreated, lipofectamine-treated, or SSO-0202-treated cells (Fig. 6C). We also confirmed that the SSOs from the downstream intron did not affect the splicing of *NEDD4Le13* in any of the cell lines tested suggesting that the predictions made were highly accurate (Appendix Fig. S5). In summary, AI/ML successfully identified functional SSO binding sites, reducing the need for time-consuming microwalks for SSO optimization.

## SSOs targeted to NEDD4Le13 promoted exon inclusion and affected the TGFβ pathway

NEDD4L is a ubiquitin ligase that has been previously shown to play a role in TGFβ regulation (Gao et al, 2009; Aragón et al, 2012). Moreover, NEDD4L has been shown to be involved in both oncogenesis and tumor suppression (Xie et al, 2021; Guo et al, 2022). However, it is unclear if a specific AS isoform of NEDD4L is responsible for regulating TGFβ signaling in TNBC tumors, where it has the potential to drive multiple aspects of tumor progression. The TGFβ-dependent response is highly contextual throughout development, across different tissues, and therefore its dysregulation is highly relevant in tumor development and progression (Bellomo et al, 2016; Massagué, 2008).

Using MDA-MB-231 as a representative TNBC cell line, and MCF10A (cell line derived from breast fibroadenoma) as a control, we evaluated the splicing patterns of *NEDD4Le13* in the context of TGFβ stimulation via treatment with human recombinant TGFβ (hrTGFβ). Untreated MDA-MB-231 cells predominantly expressed the skipping isoform. When SSO-0205 was added to the cells for 24 h, there was a significant increase in *NEDD4Le13* inclusion. SSO-0205 maintained a high level of *NEDD4Le13* inclusion even after exposure to hrTGFβ for 3 or 6 h. On the other hand, even though MCF10A cells often skip *NEDD4Le13* in untreated conditions, they did not show a response to SSO-0205 under TGFβ treatment, perhaps due to differences in *NEDD4Le13* regulation between the cell types and a dependency on the short isoform in the context of tumor maintenance (Fig. 7A). It is well established that dynamics of RBP interactions vary between different cell lines and cellular conditions (Sun et al, 2021). Importantly, many RBPs are known to be multi-functional and behave differently in healthy vs tumorigenic conditions depending on the differential stoichiometry of the proteins (Zhou et al, 2023; Qin et al, 2020). Accordingly, RNA-seq analysis showed *HNRNPL* and *SRSF7* to be highly expressed in MDA-MB-231 cells compared to MCF10A cells, pointing to a specific role of this regulatory network in the context of TNBC (Fig. 7B).

Next, we evaluated the effect of SSO-0205-induced *NEDD4Le13* inclusion on the protein level and localization of several members of the TGFβ pathway. MDA-MB-231 and MCF10A cells were treated with either SSO-0205, SSO-0202 (used as a control), or with lipofectamine alone as vehicle control for 24 h followed by a 0-, 1-, or 3-h treatment with hrTGFβ. Following SSO-0205 and hrTGFβ treatment, MDA-MB-231 cells showed an accumulation of phosphorylated SMAD2/3 (phSMAD2/3) (Fig. EV4A) and TGFβ-receptor (TGFβRI) in the cytoplasm (Fig. EV4C), concomitant with a nuclear decrease of phSMAD2/3, total SMAD2/3 (Fig. EV4A) and a decrease in TGFβRI (Fig. EV4C) on the membrane compared to the lipofectamine control (LFC). These cellular localization changes were time-dependent and were prominent at 3 h after the hrTGFβ treatment. Western blot quantifications are shown in Fig. EV4E, where the effect of SSO-0205 on subcellular localization for SMAD2/3; phSMAD2/3 and TGFβRI is highlighted. SSO-0202 treatment on MDA-MB-231 didn't promote relevant subcellular localization changes compared to the lipofectamine control group (Fig. EV4A,C,E). MCF10A cells showed very modest effects on TGFβ pathway members' subcellular localization (Fig. EV4B,D) owing to the lack of response to the SSOs by MCF10A cells described above with <20% inclusion ratio upon SSO treatment (Fig. 6).

NEDD4L has been previously described to interact with phSMAD2/3, SMAD7, and TGFβRI as part of its role in the ubiquitination and subsequent proteasomal degradation of these proteins (Gao et al, 2009; Kuratomi et al, 2005). We used MG132 to inhibit proteasome activity and performed immunoprecipitation assays for SMAD2/3 and TGFβRI to determine their interaction with NEDD4L after treating cells with SSO-0205 for 24 h, then with hrTGFβ for 0, 3, or 6 h. When we inhibited proteasome activity before treating with SSO-0205, the total levels of NEDD4L, phSMAD2/3, and TGFβRI increased after hrTGFβ addition to the cells at 3 and 6 h, specifically in the MDA-MB-231 cells (Appendix Fig. S6A top panel). Moreover, we found that SSO-0205 treatment promoted protein–protein interaction (PPI) between NEDD4L and SMAD2/3 as well as between NEDD4L and SMAD7, particularly after 3 h of hrTGFβ treatment as observed by an increase of NEDD4L signal after SMAD2/3 IP (Appendix Fig. S6B) and SMAD7 IP (Appendix Fig. S6C). Importantly, MCF10A cells treated with SSO-0205 and hrTGFβ did not show any change in downstream TGFβ signaling protein subcellular localization (Appendix Fig. S6A lower panel) or PPIs by IP (Appendix Fig. S6B,C). These subcellular fractionation data and PPI results confirm the direct role of the NEDD4L inclusion isoform in modulating TGFβ pathway activity by marking the phSMADs and TGFβRI for degradation as previously described (Gao et al, 2009; Kuratomi et al, 2005).

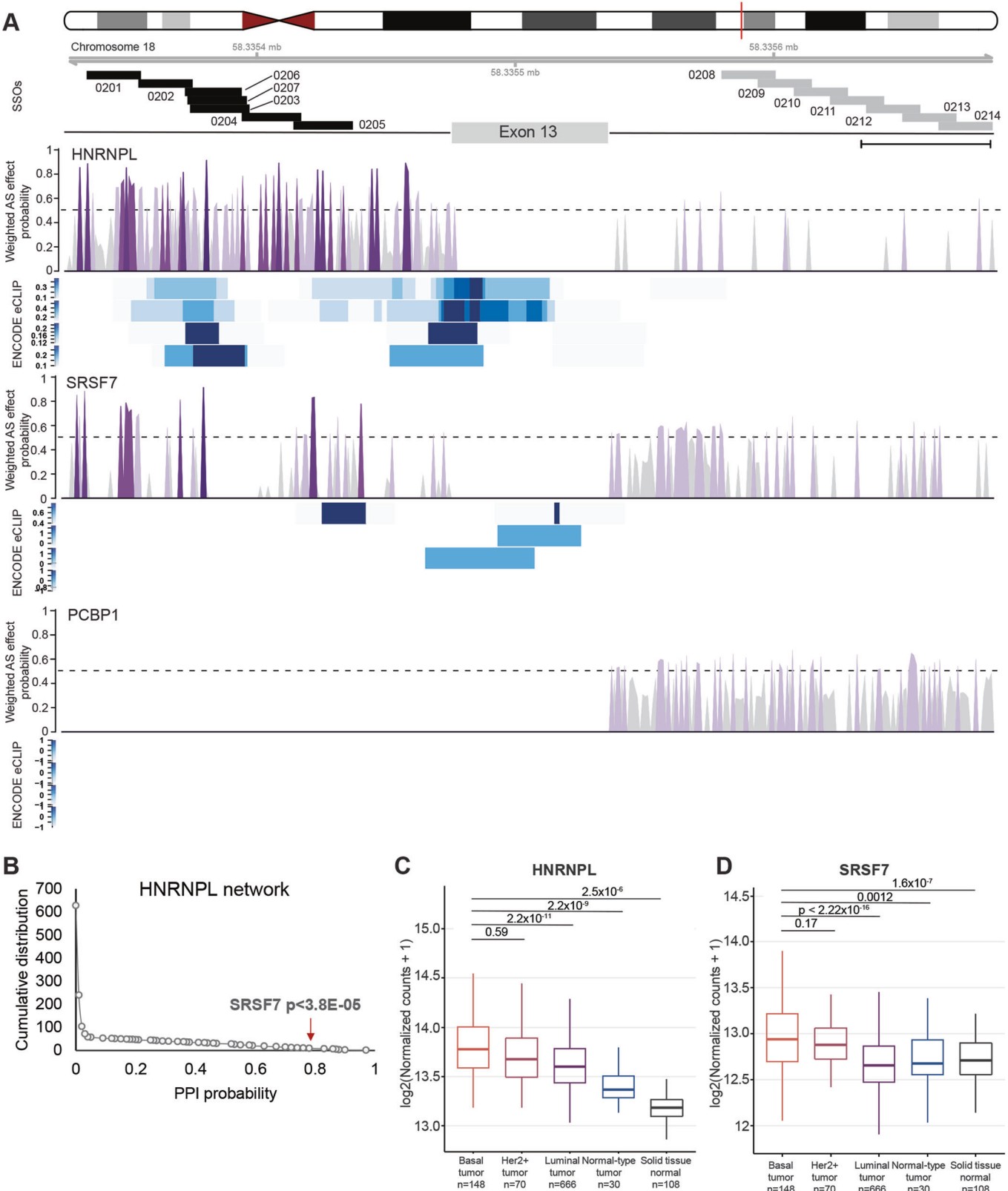

**Figure 5.  XGboost model predicts the optimal functional SSO binding site to modulate the splicing of NEDD4Le13 in TNBC.**

(A) SF-specific AS effect probability scores as determined by SHAP analysis. PCBP1 is included as a negative example. Black boxes in the top track indicate the binding sites of the SSOs designed and tested to promote *NEDD4Le13* exon inclusion. Gray boxes in the top track indicate the positions of SSOs not predicted to regulate exon inclusion, used as negative controls. Blue heatmaps indicate eCLIP binding from ENCODE cell lines, whereas darker blue indicates a higher binding score. Scale bar: 50 nucleotides. (B) Probability distribution of protein–protein interaction scores between *HNRNPL* and the rest of the SFs in the Akerman et al, 2015 dataset. *P* value and probability are highlighted for *SRSF7*. As described by Akerman et al, 2015, the probabilities are predicted using a Bayesian probability model. The *p* value was calculated using a Mann–Whitney test. (C,D) *HNRNPL* and *SRSF7* expression levels across the subtypes of breast cancer in TCGA. Basal tumor samples $n = 148$, Her2+ tumor samples $n = 70$, Luminal tumor samples $n = 666$, Normal-type tumor samples $n = 30$, Solid tissue normal samples $n = 108$. *P* values were calculated using a Wilcox test. The box plot center line indicates the median value, upper and lower box edges correspond to the 25th and 75th percentiles, and the whiskers extend to 1.5 * interquartile range from the box edge.

## SSO-mediated NEDD4L exon inclusion selectively modulated TNBC cell proliferation

To understand the role of the NEDD4Le13 in TNBC cell viability, we conducted a cell titer glow viability assay in the presence or absence of SSO-0205. Treatment of MDA-MB-231 cells with SSO-0205 significantly decreased the viable cell count in MDA-MB-231 cells both in fixed concentration (400 nM), as well as in a dose-dependent manner. (Fig. 7C; Appendix Fig. S6B). Additionally, a cell cycle evaluation indicated that SSO-0205 treatment significantly increased the number of cells in G1 and decreased the number of cells in G2 at the 24-h time point (Fig. 7D). Treatment of MCF10A with SSO-0205 did not change cell proportions along the cell cycle or cause a significant change in cell viability (Fig. 7C,D). To determine whether NEDD4Le13 AS modulates the cell cycle in response to TGFβ, we analyzed the gene expression of *CDKN1A* and *c-MYC*, two key downstream targets of TGFβ in epithelial cells (Decker et al, 2021; Weiss, 2003). We observed that 24 h after SSO-0205 treatment, and prior to hrTGFβ treatment, MDA-MB-231 cells showed a decrease in both CDKN1A and c-MYC gene expression levels compared to the control cells treated with lipofectamine (Fig. 7E). Interestingly, when hrTGFβ was added to the cells, the increase in CDKN1A gene expression observed in the lipofectamine control group was inhibited by the SSO-0205 addition in the SSO treatment group. (Fig. 7E) (TGFβ time 3 and 6 h). Additionally, we saw that *SMAD7* expression showed a similar expression profile to the *CDKN1A* when comparing the lipofectamine control to SSO-0205 treated cells (Fig. 7E). Consistent with the viability assay, these changes were limited to MDA-MB-231 cells and not observed in MCF10A cells. Taken together, these data indicate that promoting *NEDD4Le13* inclusion with SSO-0205 specifically modulated the proliferation and viability of the MDA-MB-231 cells, further supporting its biological importance and therapeutic potential in TNBC.

## NEDD4Le13 AS modulated the MDA-MB-231 migratory response to TGFβ pathway activation

Cell migration and invasion are hallmarks of cancer that enable metastasis. *ANGPTL4, ITGA5,* and *ITGB3* are TGFβ-dependent cell migration-associated markers. SSO-0205 pre-treatment of MDA-MB-231 cells decreased the gene expression response of *ANGPTL4, ITGA5,* and *ITGB3* after hrTGFβ addition at time 0, 3, and 6 h, while SSO-0202 treatment did not cause significant changes for any of the time points (Fig. 7F). Expression changes in migration-associated genes were not observed in MCF10A cells after SSO-0205 or SSO-0202 treatment, even though TGFβ stimulation

induced the expression of some of these genes (Fig. 7F). Finally, evaluation of changes in the migration of MDA-MB-231 cells using a Transwell Migration Assay showed a 40-50% decrease in cell migration in SSO-0205 treated cells, even in the presence of hrTGFβ. This effect was specific to MDA-MB-231 cells; MCF10A cells did not show a decrease in migration in the presence of SSO-0205 (Fig. 7G).

Overall, we have shown that *NEDD4Le13* skipping contributes to TNBC tumor progression and promotes overactivation of the TGFβ pathway in MDA-MB-231 cells. Furthermore, SSO-0205 was able to reverse *NEDD4Le13* exon skipping by presumably blocking the binding of HNRNPL and SRSF7 to the intron upstream *NEDD4Le13*, thereby decreasing both the proliferative and migratory behavior of MDA-MB-231 cells. Both the target, *NEDD4Le13*, and the binding site for the compound, SSO-0205, were discovered de-novo by applying AI/ML algorithms to RNA-seq datasets, demonstrating the potential for innovative drug discovery with the SpliceCore platform in the field of RNA therapeutics.

## Discussion

Our goal for this study was to showcase the value of AI/ML in developing new SSO compounds that modulate AS. We experimentally validated this approach using *NEDD4Le13*, a new target for TNBC also predicted using AI/ML with our proprietary platform, SpliceCore.

Previous studies using AI/ML to study AS perturbations have shown that while predictive models are usually optimized for maximum performance, they often lack explainability, making it harder for scientists to make actionable decisions based on the AI/ML algorithm's results. Reduced explainability can stem from the use of complex predictive features with hidden associations, or unbalanced sensitivity/specificity, often overlooked in favor of maximizing performance metrics such as AUC (Azodi et al, 2020; Johansson et al, 2011). In contrast, tree-based learning methods, like XGboost, are inherently more explainable (e.g., when compared to deep learning) because they enable the investigation of the content and relationship between predictive features. In addition, the use of SHAP/OOB as a layer of explainability to XGboost allowed for the identification of regulatory SFs at each nucleotide of interest, which is crucial for downstream studies on the regulation of the AS event and the mechanisms by which the SSOs modulate it (Fig. 5; Appendix Fig. S1). While many factors can contribute to the success of an SSO in regulating splicing outcomes, including SF binding and RNA secondary structure, it

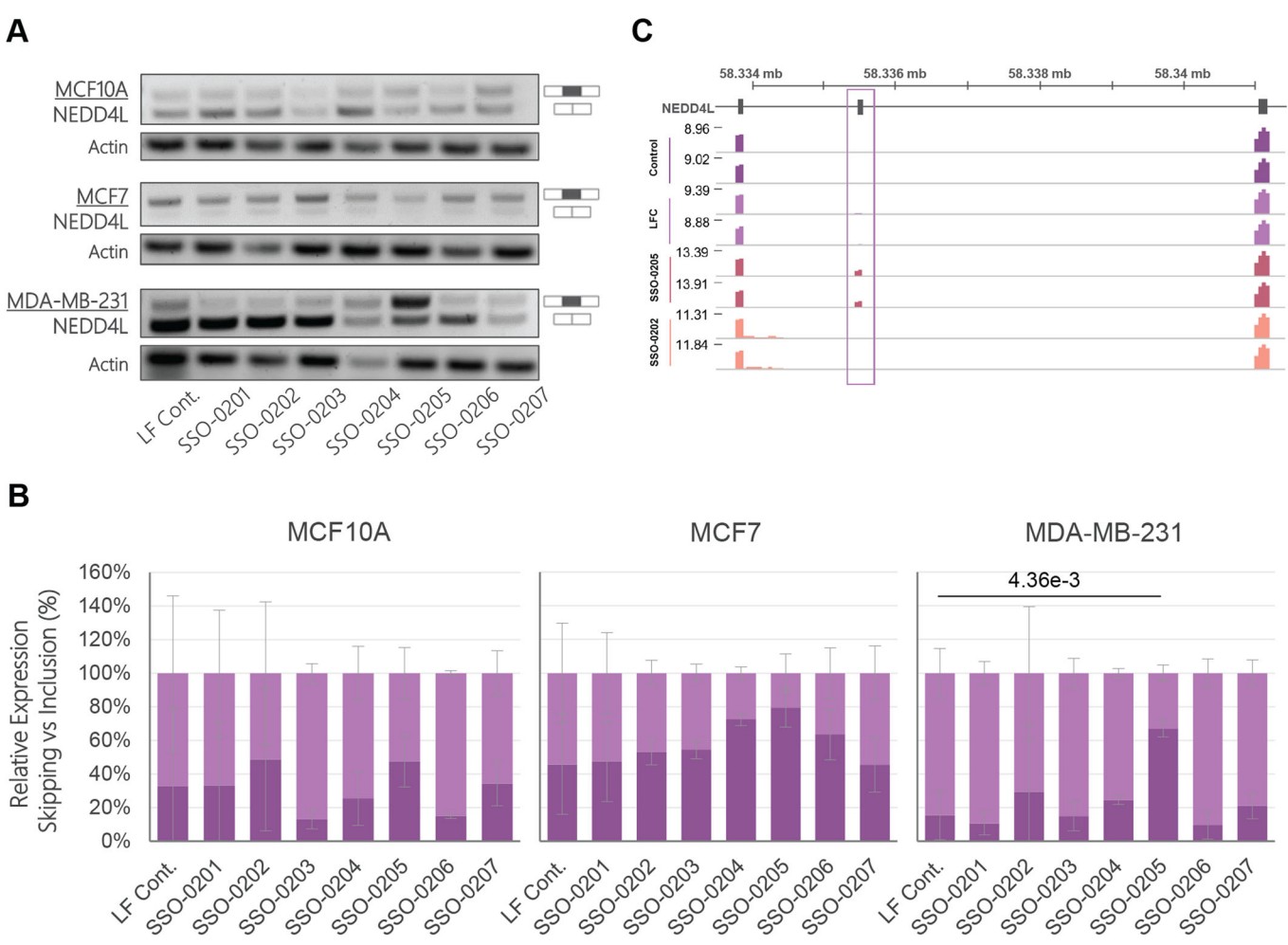

**Figure 6. Predicted SSOs modulate *NEDD4Le13* inclusion.**

(A) Representative 2% agarose gel for *NEDD4L* PCR showing isoform changes in three breast tissue cell lines treated with the indicated SSOs (400 nM) for 48 h. Actin expression is used as cDNA internal control. (B) Inclusion/skipping percentage measured by qPCR in three breast cancer cell lines after SSO (400 nM) treatment for 48 h ($n = 3$ biological replicates). Mean and standard deviation are represented. Statistical differences were calculated by Student's $t$-test vs the corresponding time point at the control (Ctr) group. (C) Genome browser tracks displaying RNA-seq data in MDA-MB-231 cells (untreated (dark purple), treated with a lipofectamine control (light purple), SSO-0205 (maroon), or SSO-0202 (peach)). Box highlights *NEDD4Le13*. Source data are available online for this figure.

has been shown that SFs preferentially bind single-stranded RNA and that SFs have stronger effects on splicing when bound to linear RNA (with no secondary structure) (Hiller et al, 2007). Clearly, the incorporation of SF motifs and interaction data is sufficient to predict whether an SSO will be effective. There have been other ML approaches developed for SSO selection, such as eSkip-Finder (Chiba et al, 2021; Zhu et al, 2023), however, this tool exclusively identifies exonic SSOs that promote exon skipping and is based solely on sequence features. Exon inclusion is often regulated by SF binding in the upstream and downstream introns, and SSOs can be more specific when binding to intronic regions. Therefore, utilizing features based on splicing regulatory information, SF binding characteristics, and sequence information allows for a more comprehensive evaluation of SSO effectiveness, and, importantly, allows for the identification of robust SSOs that promote either exon inclusion or exclusion. Ultimately, our AI/ML approach eliminates the need for time-consuming and expensive SSO microwalks that require 100–200 tiling oligos to be tested manually

for activity. AI/ML-based SSO design not only consolidates the number of oligos tested, but also informs the potential regulatory mechanism associated with the targeting sites that can be tested with experimental methods such as overexpression of knockdown of corresponding SFs.

Remarkably, the prospective validation study presented here produced significant biological insights related to splicing regulation in TNBC. First, we identified seven AS events highly recurrent in TNBC. Of particular interest, AS of *NEDD4Le13* was found to be present in 64% of TNBC patient tumors. Second, we showed that *NEDD4Le13* is a promising target for TNBC since splice-switching with SSO-0205 caused TNBC-specific viability loss (Fig. 7). This effect may be partially explained through altered localization of SMAD proteins downstream of pro-tumorigenic TGFβ activation (Appendix Fig. S6). The tumor-specific anti-proliferative effects of SSO-0205 are encouraging and may result in reduced toxicity in future pre-clinical and clinical development compared to other drugs targeting critical cellular factors. Third, skipping of

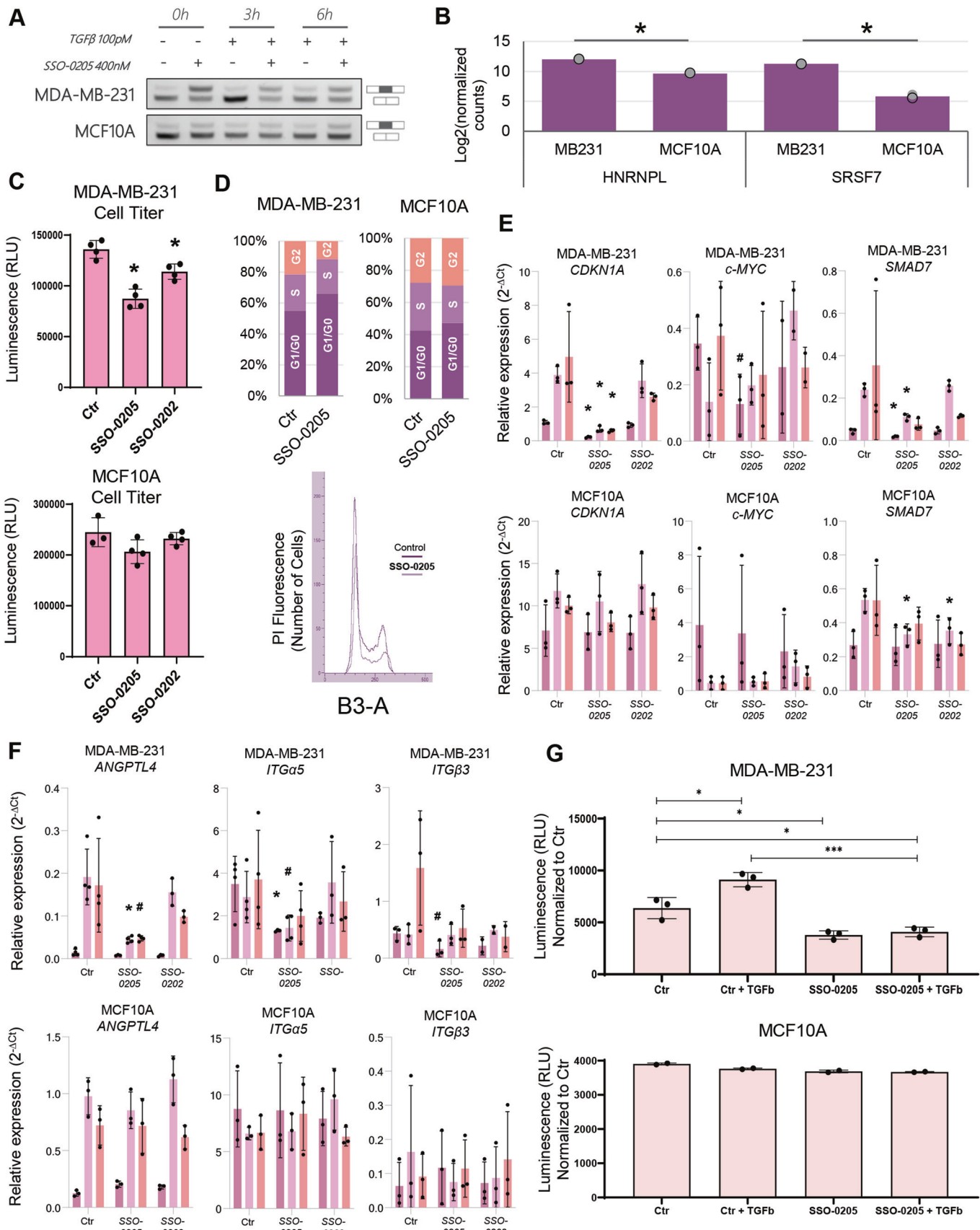

**Figure 7. SSO-0205 modulates *NEDD4Le13* inclusion, causes cancer cell migratory response to TGFβ, and decreases cell proliferation.**

(A) PCR measuring *NEDD4L* isoforms in MDA-MB-231 and MCF10A cells in response to TGFβ stimulation (0, 3, or 6 h) after SSO-0205 treatment (400 nM) (24 h). (B) Normalized RNA-seq counts for HNRNPL and SRSF7 in MCF10A and MDA-MB-231 cells showing significantly higher expression of these SFs in MDA-MB-231 cells. Statistical differences calculated with DESeq2; *≤1 × 10$^{-85}$ ($n = 2$ biological replicates). (C) Cell proliferation as measured by Cell TiterGlo® 96-well assay. MDA-MB-231 (top graph) and MCF10A (bottom graph) cells were treated with lipofectamine alone as a control or with the indicated SSO for 48 h (400 nM) ($n = 4$ biological replicates). (D) Cell cycle phase analysis measured by propidium iodide flow cytometry (G1/G0; S phase and G2). The cell cycle phase for each cell line is represented by % (top graphs). One-way ANOVA analysis was performed to compare differences between control and SSO-0205 treatment, MDA-MB-231 $p$ value = 0.0018, MCF10A $p$ value = 0.1596. A representative Propidium Iodide histogram plot for MDA-MB-231 is shown in the lower panel. Cells were treated with lipofectamine alone or with SSO-0205 (400 nM) for 24 h ($n = 3$ biological replicates). (E) qPCR quantifying expression levels of the indicated cell cycle-related genes in MDA-MB-231 (top line) or MCF10A (bottom line) cells treated with SSO or lipofectamine alone as a control for 24 h followed by 0, 3, or 6 h treatment with hrTGFβ ($n = 3–4$ biological replicates). (F) qPCR quantifying expression levels of the indicated migration/invasion-related genes in MDA-MB-231 (top line) or MCF10A (bottom line) cells treated with SSO (400 nM) or lipofectamine as a control for 24 h followed by 0, 3, or 6 h treatment with hrTGFβ ($n = 3–4$ biological replicates). (G) Transwell migration assay on the indicated cell types (MDA-MB-231 top graph and MCF10A bottom graph), cells were treated with lipofectamine or SSO-0205 (1 μM) for 24 h, followed by overnight TGFβ treatment in the Transwell chamber ($n = 2–3$ biological replicates). Mean and Standard deviation are represented in C, E–G. Statistical differences were calculated by Student's $t$-test vs the corresponding time point at the control (Ctr) group; *≤0.05; **≤0.01; ***≤0.001; #≤0.1. Source data are available online for this figure.

NEDD4Le13 causes the exclusion of a protein loop region between the WW1 and WW2 domains important for protein interaction (Fig. 4B,C) (Gao et al, 2009; Aragón et al, 2012). This loop region contained a threonine residue that can be phosphorylated, potentially adding an additional regulation through post-translational modifications (Snyder et al, 2004). Fourth, SSO-0205 treatment affected TGFβ-dependent gene expression changes in proliferation- (*CDKN1A, C-MYC*) and invasion-related genes (*Integrins α5 and β3, ANGPTL4*), leading to significant changes in cell cycle and cell migration (Fig. 7). While the TGFβ-independent downregulation of *CDKN1A* in the presence of SSO-0205 was surprising, it is possible that additional players modulate *CDKN1A* levels through non-canonical TGFβ regulation (Weiss, 2003). Finally, despite the abundance of *NEDD4Le13* skipping in MCF10A cells (a non-cancerous mammary fibroadenoma cell line), SSO-0205 did not promote exon inclusion in these cells, suggesting that the TNBC-specific SF network identified hereby could have a different role augmenting pro- tumorigenic activity, further supported by the observed TNBC-specific overexpression of *HNRNPL* and *SRSF7*. We suggest that *NEDD4Le13* skipping is lineage-restricted to basal origin cell lines like MCF10A and MDA-MB-231; however, AS regulation appears to divert between both cell types, such that exon skipping in the latter promotes tumor progression, which may lead to dependency for *NEDD4Le13* skipping in the context of TNBC (Fig. 7B). However, it is also plausible that there may be additional regulatory factors in MCF10A which may tightly regulate the abundance of each isoform in a non-cancerous condition. Additional work will elucidate the differential effect of HNRNPL and SRSF7 in normal and malignant cell lines in the NEDD4Le13 context.

Currently, several SSOs remain under investigation as potential treatments for cancer and other diseases, providing an alternative to small molecules, which are currently limited to a subset of "druggable" proteins (Havens and Hastings, 2016; Xiong et al, 2021). There is still progress to be made in the field of antisense therapeutics, especially in the area of drug delivery, but this is a highly active area of research and is the focus of several biotechnology companies (Roberts et al, 2020). Ultimately, this study illustrates the usefulness of an innovative AI/ML algorithm developed upon key principles of RNA biology, to extract novel, actionable insights from RNA-seq data. We have shown that it is possible to identify a novel drug target, and design an effective SSO against it, that not only promotes exon inclusion but also displays

anti-cancer activity. The usefulness of AI/ML in extracting value from RNA-seq data to identify and drug novel targets can be expanded beyond cancer research into other diseases driven by splicing alterations, like neurodegeneration or metabolic diseases (Syed, 2016; Finkel et al, 2017; Han et al, 2020; Kim et al, 2019; Centa et al, 2020; Wagner et al, 2021).

# Methods

## The SpliceCore platform

The SpliceCore software platform is Envisagenics' proprietary technology for the discovery of splicing-modulatory drug targets using RNA-seq data. SpliceCore utilizes scalable cloud computing through Microsoft Azure services to efficiently perform *de-novo* transcript assembly followed by AS analysis and biological explainability. SpliceCore implements the following algorithms: SpliceTrap, a Bayesian-based method for RNA-seq alignment and AS quantification, and SpliceDuo, a regression-based predictive model for AS cross-comparison. (Wu et al, 2011; Anczuków et al, 2012) (Appendix Fig. S3; Appendix Supplementary Methods). SpliceCore performs de-novo transcript assembly using an exon-centric reference transcriptome called TXdb (Wu et al, 2011). Exon-centric analysis differs from transcript-centric analysis in that it treats the transcriptome as a collection of independent AS events rather than full-length transcripts (Appendix Fig. S4; Appendix Supplementary Methods).

## XGboost model development

### SFs used for feature engineering

SF binding motifs were derived from three independent methods. The first was RNACompete, an in vitro binding enrichment approach to identify SF binding preferences using libraries of random k-mers and quantification using microarrays (Ray et al, 2013). RNA-compete enrichment scores (E-scores) covering all possible RNA 7-mers for 85 SFs were downloaded from the CISBP (http://cisbp-rna.ccbr.utoronto.ca/). Second was Bind-n-Seq, which is similar to RNACompete and uses RNA-seq instead of microarray (Lambert et al, 2014). Bind-n-Seq E-scores covering all RNA 7-mers for 58 SFs were downloaded from ENCODE (https://www.encodeproject.org/publication-data/ENCSR876DCD/). Finally, we used RBPmap, a

computational method for the prediction and mapping of SF binding sites (Paz et al, 2014). RBPmap screens pre-mRNA sequences with the position-specific scoring matrixes (PSSMs) of 97 SFs, based on the weighted-rank algorithm, which considers the clustering propensity of PSSMs and the overall tendency of the regulatory region to be conserved. In total, 205 SFs were covered by at least one of these three methods, including 35 known AS regulators, 17 core spliceosomal proteins, and 142 spliceosome-associated proteins (Akerman et al, 2015).

### Differential SF binding estimation

Every SNV was paired with its reference allele sequence to estimate the change in binding score of 205 SFs according to $\triangle binding = S(snv, |, SF, method) - S(ref, |, SF, method)$, where "S" represents E-score (for RNA-compete and bind-n-Seq) or binding probability (for RBPmap) of a binding motif determined by a SF analysed with a given method. The absolute $\triangle binding$ scores were divided into ten quantiles and multiplied by their sign (1 or $-1$), such that $Q_{sf,method} = q(|\Delta binding|) * \text{sgn}(\Delta binding) < 0$ informs increased binding score, $Q_{sf,method} = q(|\Delta binding|) * \text{sgn}(\Delta binding) > 0$ is a decrease in binding score and $Q_{sf,method} = q(|\Delta binding|) * \text{sgn}(\Delta binding) \sim 0$ signifies that the no effect on SF binding.

### Splicing factor clusters (SFCs) compilation

To reduce the sparsity of the feature space while introducing biological meaning into the model, we sorted individual $Q_{sf,method}$ scores into 83 SFCs based on their probability to physically interact and perform similar functions. In brief, we first assigned SFs to 32 spliceosomal ontologies describing various aspects of spliceosome assembly and function. These ontologies, in turn, belong to six different functional classes: (1) AS regulation (e.g., activators and repressors), (2) core spliceosomal subcomplexes (e.g., U2 snRNP), (3) binding motifs (e.g., SFs that prefer binding UG-rich motifs), (4) Tissue specificity (ranked into four quartiles), (5) evolutionary conservation (e.g., human vs. mice or yeast) and (6) others (unassigned spliceosomal proteins). AS regulators, core spliceosomal subcomplexes and "others" were derived from annotations in (Akerman et al, 2015). Nucleotide content of binding motifs was estimated based on SF binding profiles in (Ray et al, 2013; Lambert et al, 2014; Paz et al, 2014). To compute tissue specificity quantiles, we downloaded mRNA expression profiles from the GTEx website (Lonsdale et al, 2013) and estimated the tissue specificity index as described in (Yanai et al, 2005) and divided SFs into four tissue specificity quartiles. Human/mouse and human/yeast evolutionary conservation SF status were derived from Homologene (Cui et al, 2007). Finally, essential gene status was derived from the DEG database (Zhang et al, 2004). See Dataset EV1 for the SF composition of each SFC.

For each protein pair, we estimated their probability to physically interact in the context of a given ontology. For this, we used the "probabilistic spliceosome", a protein network model based on yeast-two-hybrid and microarrays gene expression data to study spliceosomal dynamics and regulation (Akerman et al, 2015). Finally, to generate SCs we applied cluster analysis to the resulting protein–protein interaction probability matrixes. We applied hierarchical agglomerative clustering, using the Unweighted Pair-Group Method with Arithmetic Mean (UPGMA) for calculating distances between SCs whereby a dendrogram was built to

represent a similarity matrix, and the nearest two clusters were iteratively combined into a larger cluster (using Scipy 1.5.2 with Python 3.6). Since SFs can exist in more than one SC, and each SC contains variable number of SFs, it is possible for a given SNV to show $|Q_{sf,method}| > 0$ for more than one SF per SFC. For simplicity, it was assumed that the SF with the highest scoring $|Q_{sf,method}|$ is the one occupying and binding that SNV. Ties for two or more SFs scoring equally high were resolved using SHAP values.

### Machine Learning software and models

All machine learning models and OOB analysis used in this project were implemented using the open-source Scikit learn 0.22 (https://scikit-learn.org/stable). SHAP analysis was performed using the SHAP python package (version 0.39.0) (https://shap.readthedocs.io/en/latest/). The models tested were evaluated by three metrics, AUC, sensitivity, and specificity. AUC, or area under the receiving operating characteristic curve, measures the performance of a model by determining how much the model can differentiate between positive and negative classes. Sensitivity and specificity measure the accuracy of a model, where sensitivity is the true positive rate and specificity is the true negative rate.

### Cell lines

All cell lines used in this study were purchased from NCI Cancer Cell line-60 panel and/or American Type Culture Collection (ATCC) authenticated by STR profiling and mycoplasma testing.

### RNA-sequencing

Approximately $5 \times 10^5$ cells were harvested in biological triplicates from each breast cancer cell line (MCF7, T47D, MDA-MB-231, MDA-MB-468, HS578T, BT549, and MCF10A). Frozen cell pellets were provided to GeneWiz (SD, California) for RNA extraction and library preparation. Briefly, 1 ug of RNA extracted using TriZol was enriched for PolyA-containing RNA using oligo-dT columns. Libraries for polyA+ RNA-seq were prepared using TruSeq chemistry (Illumina), multiplexed, and sequenced to obtain paired-end 101-base-pair (bp) reads on an Illumina HiSeq 2000 platform, resulting in 30 million to 45 million reads per library.

### NEDD4L protein structure modeling

The three-dimensional model of the NEDD4L amino acids 193–418 was obtained by ab initio modeling on the Robetta platform (http://robetta.bakerlab.org/, D. E. Kim et al, 2004). Five models were obtained and all of them were validated in the Molprobity server (http://molprobity.biochem.duke.edu/, Davis et al, 2007). Two energy minimization processes using the Chimera program (Yang et al, 2012) were followed by residue-specific minimization in SPDBV (Guex and Peitsch, 1997). The model with the best molprobity score was used for comparative modeling of alternate structures with exclusion. Root mean square deviation was calculated by superimposing two structures in PYMOL.

### RNA extraction, PCR, and quantitative PCR

Total RNA from different breast cancer cell lines were extracted using the TriZol reagent as per the manufacturer's instruction. One microgram of RNA was used to make cDNAs using oligo-dT primers. One-tenth of the volume of the cDNA reaction mixture was used in the PCR reaction containing specific forward and reverse primers for detecting different splicing isoforms. The

**Table 1. Primer sequences.**

| Gene name | Forward primer | Reverse Primer |
| --- | --- | --- |
| NEDD4Le13 | GAACCCTCCTCAAGGTTGAGGTCAT | ATGATAGGTCGAGTCCAAGTTGTGGTTC |
| MAP3K7e11 | GTGAATCTGGACGTTTAAGCTTGGGAG | GACCAGGTTCTGTTCCAGTTACAGTCA |
| NFYAe3 | GTGTACCTCACAGCCTTCTAGGATCT | TGTGTTCCAGAAACAGGTACTTGCATGATG |
| ESYT2e14 | CCAAATGCGTCAAACCTCGA | CCTTGTGCCCAACTGACATC |
| ABI1e9 | GAAGTAGTGGAGGAAGTGGAAGTCGA | ACTACTGCAGCCTCCTCATCTTCATAATC |
| MARK2e15 | TCCTCTCCACCAGCACAAAT | AAGTTAGTTCGGTCTGGGGC |
| FLNBe31 | ATGCCGGTAACCGAGAAGGATCT | TTAAGGCACTGTGACATGAAAAGGGCT |
| ACTIN | TTTTGGCTATACCCTACTGGCA | CTGCACAGTCGTCAGCATATC |
| CDKN1A | ATGAAATTCACCCCCTTTCC | CCCTAGGCTGTGCTCACTTC |
| c-MYC | CTGGGAAGAAGCCAGTTCAG | TGGGCCATAGGTTTTCAGAG |
| SMAD7 | CTTCATCAAGTCCGCCACAC | CTGATCTGCACGGTAAAGCC |
| ANGPTL4 | GGGTCTGGAGAAGGTGCATA | GTGGAGAAGGGTACGGAGAG |
| ITGα5 | TTCTGTAGCTGCCACTGACA | CAAACCGTGCAAAGACCTCA |
| ITGβ3 | TGGGGCTGATGACTGAGAAG | ACGCACTTCCAGCTCTACTT |
| GAPDH | TGTGGGCATCAATGGATTTGG | ACACCATGTATTCCGGGTCAAT |

amplicons are separated in 2% agarose gels. For qPCR Universal SYBR Green Supermix (BioRad) manufacturer protocol was followed. GAPDH was used as housekeeping. The primer sequences for genes used in the PCR assays are in Table 1.

### Splice-switching assays

Computationally predicted SSO sequences were purchased from Microsynth Technologies, Switzerland. The oligos were uniformly modified to contain phosphothioate backbone and 2′ methoxyethane containing ribose sugars (2′ MOE). All oligos were HPLC purified and lyophilized oligos were reconstituted in calcium and magnesium-free PBS to obtain 100 uM stocks. The cells were plated to 80% confluency and 400 nM SSO specific to *NEDD4L* or control SSOs was transfected using Lipofectamine 3000 (Thermo Fisher Scientific, L3000001) according to the manufacturer's instruction. Alternatively, linear concentration of doses of the oligos were used for the dose-response assay. The cells were harvested, and RNA was extracted 48 h post-transfection. RT-PCR evaluation of the splice-switch was performed as described above. siRNA treatment was performed using commercial siRNA from Thermo Fisher Scientific; siControl (Cat. # 4390843), siHNRNPL (Cat. # 4392429), and siSRFS7 (Cat. # 4392420). Cells were plated in 12-well plates to be at 60–80% confluency at the moment of transfection. Cells were transfected using Lipofectamine 3000 following manufacturer recommendations, siRNAs were added at 20 nM final concentration. RNA and protein were harvested after 70 h of treatment.

### Cell viability assay

Cell TiterGlo® (Promega) was used to determine the viability of the SSO-transfected breast cancer cells according to the manufacturer's instructions. Briefly, about 10 K MCF7 or MDA-MB-231 cells were plated into each well of a 96-well plate that was treated with either control or single concentration or linear concentrations (100 nM, 200 nM, 400 nM, 600 nM, 800 nM, 1 uM) of *NEDD4L* specific SSOs in the presence of lipofectamine in triplicates. Forty-eight hours post-transfection equal volume of Cell TiterGlo reagent was added to the well,

and the total luminescence was measured using a plate reader (Perkin Elmer). The average percentage of viable cells that is proportional to the total luminescence was calculated and plotted to obtain viable cell percentage and dose-response curve for *NEDD4L* SSO.

### Cell culture and cell treatments for TGFβ pathway studies

MDA-MB-231 cells were grown in RPMI media containing 10% fetal bovine serum (FBS), penicillin, and streptomycin. MCF10A cells were grown in DMEM/F12 media supplemented with MEGM™ Mammary Epithelial Cell Growth Medium SingleQuots™ Kit (Lonza, CC-4136) and penicillin and streptomycin. For the TGFβ pathway analysis experiments, cells were plated in six-well plates at $2 \times 10^5$ cells/well. SSOs treatment were added to the cells growing in complete media at 400 nM final concentration, in the presence of Lipofectamine 3000 following manufacturer instructions, unless otherwise specified. Lipofectamine alone was added as a control group. Human recombinant TGFβ (hrTGFβ) (R&D Systems), was administered directly into the media at 100 pM final concentration. RNA or protein samples were harvested at different time points. Nuclear, cytoplasm, and membrane protein fractions were extracted following the manufacturer's instructions for the Subcellular Protein Fractionation Kit for Culture Cells (Thermo Scientific, Cat. # 78840).

### Cell cycle analysis

MDA-MB-231 and MCF10A cells were plated on six-well plates at $1 \times 10^5$ cells/mL. The following day the cells were treated with SSO-0205 at 400 nM for 24 h. Cells were trypsinized, fixed with 70% ethanol on ice, and stained with Propidium Iodide (Invitrogen, P3566) at 50 μg/mL final concentration for 30–40 min. Stained cells were analyzed by flow cytometry MACSQuant® Analyzer (MACS Miltenyi Biotech) following the manufacturer's instructions.

### Transwell migration assay

MDA-MB-231 and MCF10A cells were plated on six-well plates at $1 \times 10^5$ cells/mL, treated with SSO-0205 at 1 μM (no lipofectamine

– free uptake) for 24 h, trypsinized and resuspended in serum-free media at $5 \times 10^5$ cells/mL containing SSO-0205 at 1 µM with or without hrTGFβ 100 pM. About 500 µL of cell suspension was added to the transwell inner chamber (8-µm pore size) (Falcon, #353097), and the inserts were placed on a 12-well plate containing complete media to promote chemoattraction to the outer chamber. Cells were incubated overnight, and migratory cells were measured following the manufacturer's instruction for the cell dissociation and Calcein AM staining and luminescence measurement (Cultrex® Cell Invasion Assay, Trevigen, Cat. # 3455-024-K).

### Western blot

Total cell lysates were prepared from breast cancer cell lines using RIPA or NP-40 buffer in the presence of a protease inhibitor. Nuclear, cytoplasmic, and membrane protein extracts were obtained according to the Subcellular Protein Fractionation Kit for Culture Cells (Thermo Scientific #78840) manufacturer instructions. About 10 ug of total protein was separated in 4–15% gradient gel. The membrane-bound specific protein bands were detected using specific primary antibodies NEDD4L (Cell Signaling, Cat.# 4013 S) 1:1000, phSMAD2/3 (Cell Signaling, Cat.# 8828 S) 1:1000, SMAD2/3 (Cell Signaling, Cat.# 8685 S) 1:1000, TGFβ Receptor I (R&D Systems, Cat.# AF3025), 1:500, Ubiquitin (Cell Signaling, Cat.# 3936 S) 1:1000, Calreticulin (Cell Signaling, Cat.# 12238 S) 1:1000, Tubulin (Cell Signaling, Cat.# 2125 S) 1:1000, SRSF7 (Cell Signaling, Cat.# 82637) 1:1000, HNRNPL (Cell Signaling, Cat.# 65043) 1:1000, Actin (Sigma Aldrich, Cat.# A1978) 1:5000 and H3 (Cell Signaling, Cat.# 9717 S) 1:1000 followed by incubation with HRP conjugated secondary antibody and chemiluminescence detection (Biorad) according to the manufacturer's instruction. Tubulin and Actin were used as a loading control.

## Data availability

The datasets produced in this study are available in the following databases:

RNA-seq data: Gene Expression Omnibus GSE215917.

The trained ML models used in this study can be found on GitHub:

https://github.com/envisagenics/sso-xgboost-models/releases/tag/v1.0

The source data of this paper are collected in the following database record: biostudies:S-SCDT-10_1038-S44320-024-00034-9.

## Peer review information

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

## Acknowledgements

We are grateful to our advisors, Adrian R. Krainer, PhD (Professor & Program Chair, Cancer Center Deputy Director of Research Cold Spring Harbor Laboratory), Michael Q Zhang, PhD (Professor & Director, Center for Systems Biology University of Texas at Dallas), and Omar I. Abdel-Wahab, MD (Hematologic Oncologist, Edward P. Evans Chair in MDS, Director, Memorial Sloan Kettering Center for Hematologic Malignancies) for their guidance and fruitful discussions. Research reported in this publication was supported by the National Institute of General Medical Sciences of the National Institutes of Health under award number (R44GM116478).

## Author contributions

**Alyssa, D Fronk**: Formal analysis; Investigation; Visualization; Methodology; Writing—original draft; Writing—review and editing. **Miguel, A Manzanares**: Formal analysis; Investigation; Visualization; Methodology; Writing—original draft; Writing—review and editing. **Paulina Zheng**: Software; Formal analysis; Investigation. **Adam Geier**: Software; Formal analysis; Investigation. **Kendall Anderson**: Formal analysis; Investigation. **Shaleigh Stanton**: Formal analysis; Investigation; Visualization. **Hasan Zumrut**: Formal analysis; Investigation; Visualization. **Sakshi Gera**: Formal analysis; Investigation; Visualization. **Robin Munch**: Software; Formal analysis; Investigation. **Vanessa Frederick**: Formal analysis; Investigation; Methodology. **Priyanka Dhingra**: Formal analysis; Investigation; Methodology; Writing—original draft. **Gayatri Arun**: Conceptualization; Formal analysis; Supervision; Investigation; Visualization; Methodology; Writing—original draft; Writing—review and editing. **Martin Akerman**: Conceptualization; Software; Formal analysis; Supervision; Investigation; Visualization; Methodology; Writing—original draft; Writing—review and editing.

Source data underlying figure panels in this paper may have individual authorship assigned. Where available, figure panel/source data authorship is listed in the following database record: biostudies:S-SCDT-10_1038-S44320-024-00034-9.

## Disclosure and competing interests statement

Alyssa D Fronk, Miguel A Manzanares, Paulina Zheng, Adam Geier, Kendall Anderson, Shaleigh Stanton, Hasan Zumrut, Sakshi Gera, Robin Munch, Vanessa Frederick, Priyanka Dhingra, Gayatri Arun, and Martin Akerman were either previously or are currently employed by Envisagenics.

# Expanded View Figures

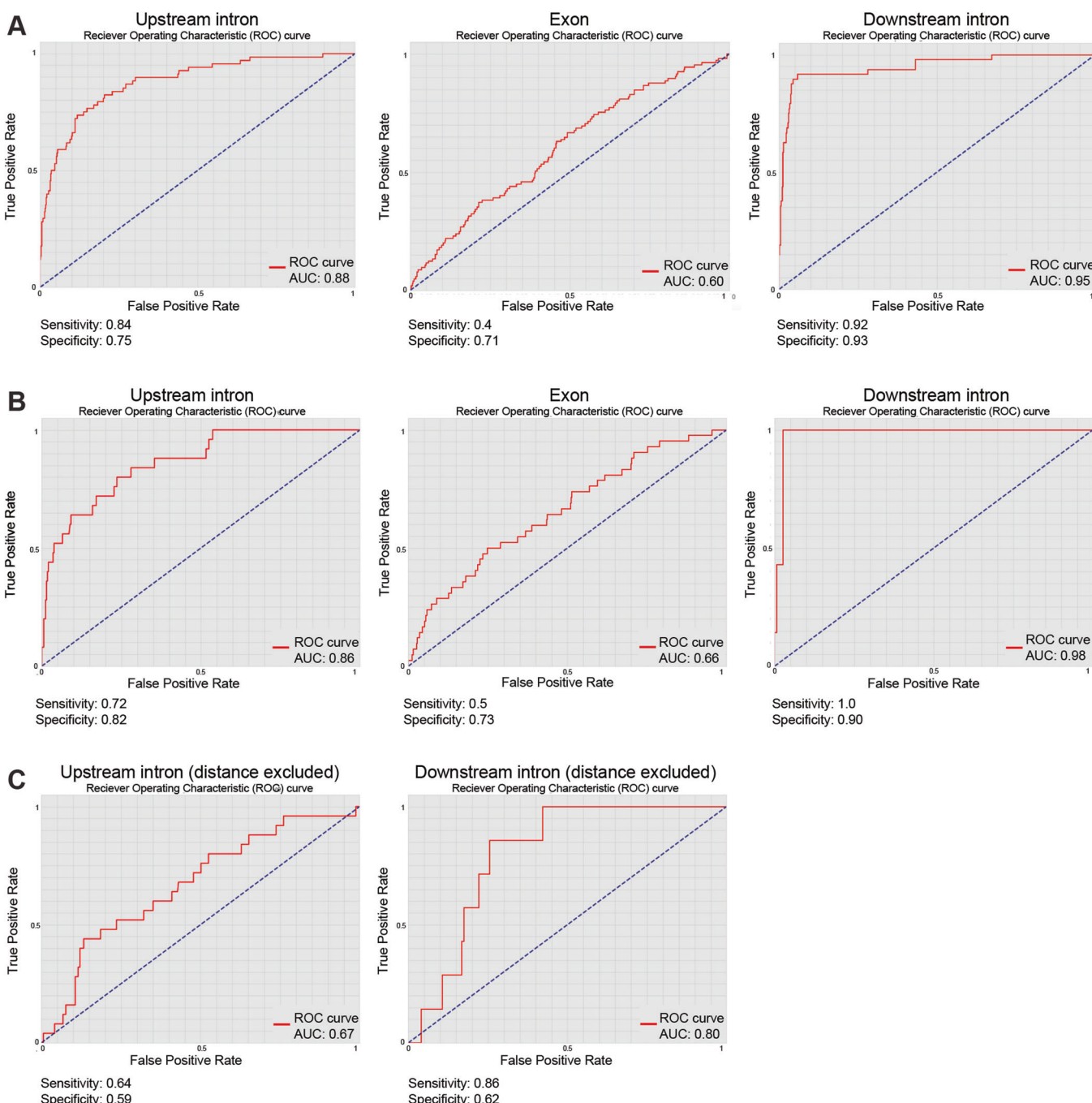

**Figure EV1.  XGboost training and testing with independent data.**

(**A**) ROC curves and sensitivity/specificity calculations for the three different models using the MFASS dataset. (**B**) ROC curves and sensitivity/specificity calculations for the three different models using the Vex-seq dataset. (**C**) ROC curves and sensitivity/specificity calculations for the XGboost-upstream and -downstream models when "distance to splice site" is not included as a feature.

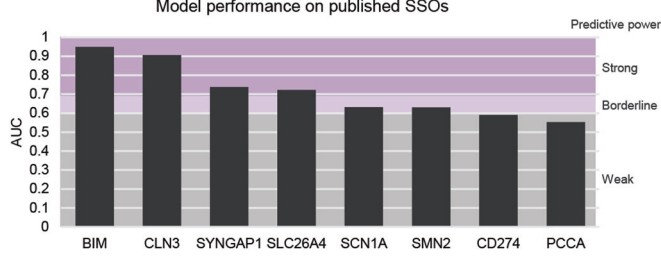

**Figure EV2. XGboost model performance on known SSO modulated AS events.**

AUC values for each alternative splicing event where both positive and negative SSOs were evaluated in previous publications. Enrichment scores were used as a metric of functional SSO prediction, and positive and negative labels were derived from the publications indicated.

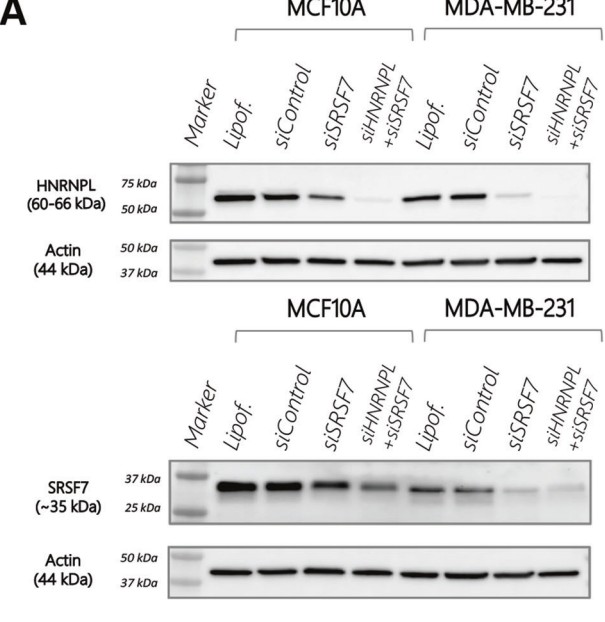

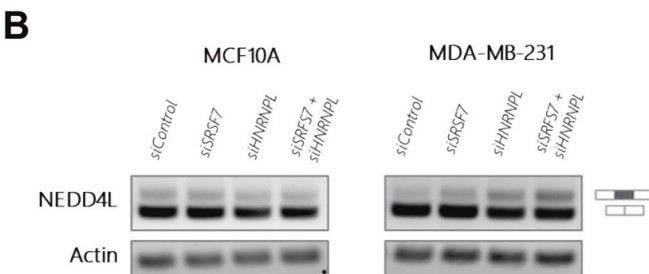

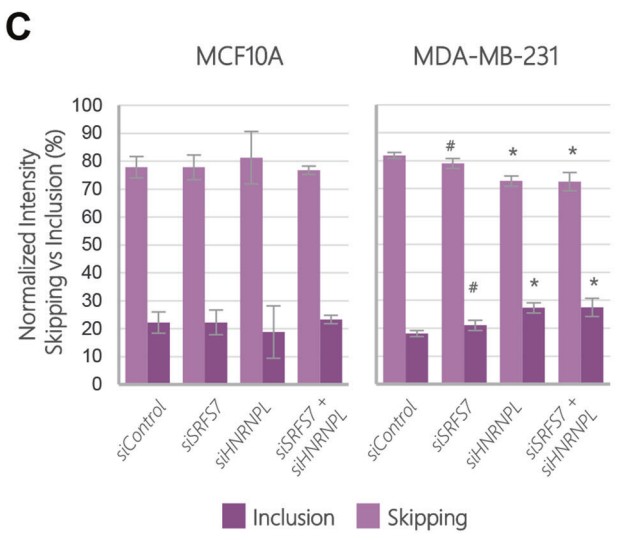

**Figure EV3.   Kock-down of the HNRNPL and SRSF7 promote NEDD4Le13 inclusion in the TNBC cancer cell line.**

(A) Western blots measuring siHNRNPL (left panel) and siSRSF7 (right panel) transfection efficiency by protein decreased in MCF10A and MB231 cells treated with corresponding siRNA alone or in combination for 70 h at 20 nM. (B) Agarose gel for PCR product measuring *NEDD4L* isoforms in MDA-MB-231 and MCF10A cells in response to siRNA treatments. (C) Quantification of agarose gels ($n = 3$ biological replicates). Mean and Standard deviation are represented. Statistical differences between each siRNA treatment group (inclusion or skipping) vs corresponding siControl were calculated by Student's $t$-test; *$\leq 0.05$; #$\leq 0.1$. Source data are available online for this figure.

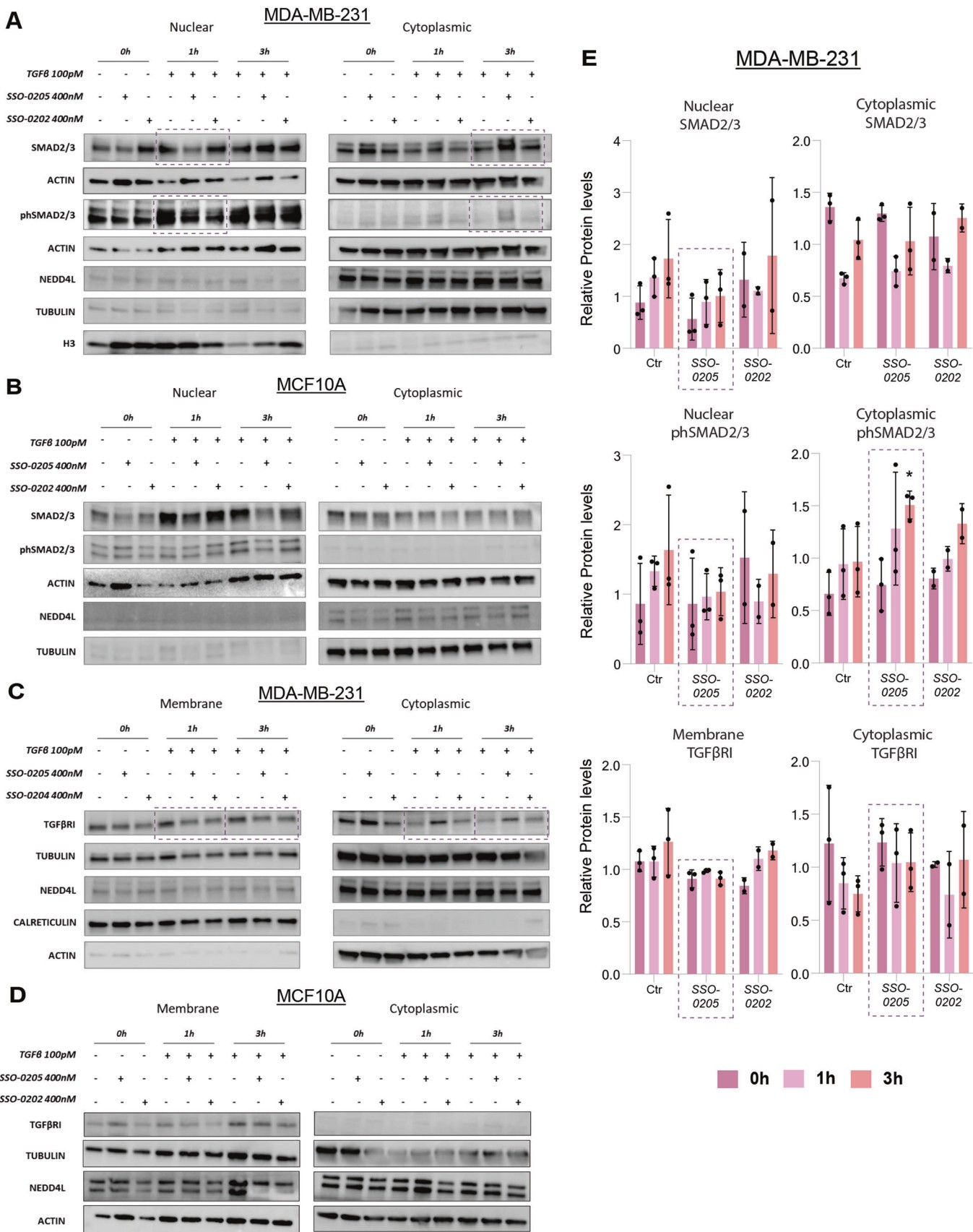

**Figure EV4.   SSO-0205 modulates the TGFβ pathway response in MDA-MB-231 cells.**

(**A,B**) Western blots measuring TGFβ-pathway related proteins in respective nuclear/cytoplasmic subcellular fractions in response to TGFβ stimulation (0, 3, or 6 h) after SSOs treatment (24 h) in MDA-MB231 (**A**) and MCF10A (**B**) Note that cytoplasmic ACTIN is shared for Cytoplasmic *SMAD2/3 and phSMAD2/3,* since the membrane was stripped and re-probed. (**C,D**) Western blots measuring TGFβ-pathway related proteins in respective membrane/cytoplasmic subcellular fractions in response to TGFβ stimulation (0, 3, or 6 h) after SSOs treatment (400 nM) (24 h) in MDA-MB-231 (**C**) and MCF10A (**D**). (**E**) Quantification of MDA-MB-231's Western blots in the respective subcellular locations for SMAD2/3 levels (top panel), phosphorylated SMAD2/3 (phSMAD2/3) levels (middle panel), and TGFβRI (lower panel). ($n = 2$–3 biological replicates). Mean and Standard deviation are represented. Statistical differences calculated by Student's *t*-test vs the corresponding time point at the TGFβ alone group; *≤0.05. Source data are available online for this figure.

