## [Peer Review File · Molecular Systems Biology]

Development and validation of AI/ML derived splice-switching oligonucleotides

Alyssa Fronk, Miguel Manzanares, Paulina Zheng, Adam Geier, Kendall Anderson, Shaleigh Stanton, Hasan Zumrut, Sakshi Gera, Robin Munch, Vanessa Frederick, Priyanka Dhingra, Gayatri Arun, and Martin Akerman

Corresponding author(s): Martin Akerman (makerman@envisagenics.com)

Review Timeline:

Submission Date:	15th Sep 23
Editorial Decision:	16th Oct 23
Revision Received:	8th Feb 24
Editorial Decision:	13th Feb 24
Revision Received:	4th Mar 24
Editorial Decision:	22nd Mar 24
Revision Received:	3rd Apr 24
Accepted:	9th Apr 24

Editor: Maria Polychronidou

Transaction Report:

16th Oct 2023

Manuscript Number: MSB-2023-12002

Title: Development and validation of an AI/ML platform for the discovery of splice-switching oligonucleotide targets

Thank you again for submitting your work to Molecular Systems Biology. We have now heard back from the two reviewers who agreed to evaluate your study. Overall, the reviewers acknowledge that the developed model is potentially relevant for future applications. However, they raise a series of concerns, which we would invite you to address in a major revision.

Without repeating all the points listed below, some of the more fundamental issues are the following:

- Reviewer #2 points out that the tool should be made publicly available, to allow its widespread use by the community. Moreover, a more detailed and transparent description of the ML model and code is required. This is an important issue, which also came up during our initial editorial evaluation of the study. According to our guidelines, data and code that is integral to the study should be made available without restrictions. Any restrictions related to the availability data, software etc. will be taken into account in the final editorial decision and may even prevent the publication of the study. Further information regarding our data availability policy can be found here: .
- Reviewer #2 mentions that further validations and control analyses are required to better support the main conclusions. They provide constructive suggestions related to these points.

All issues raised by the reviewers need to be satisfactorily addressed. As you may already know, our editorial policy allows in principle a single round of major revision, so it is essential to provide responses to the reviewers' comments that are as complete as possible. Please feel free to contact me in case you would like to discuss in further detail any of the issues raised or if you would like to share your revision plan with me. I would be happy to schedule a call.

On a more editorial level, we would ask you to address the following points:

- Please include 5 keywords.
- Please provide a .doc version of the manuscript text (including legends for the main figures) and individual production quality figure files for the main Figures (one file per figure).
- We have replaced Supplementary Information by the Expanded View (EV format). In this case, all additional figures/tables can be included in a PDF called Appendix. Appendix figures/tables should be labeled and called out as: "Appendix Figure S1, Appendix Figure S2... Appendix Table S1..." etc. Each legend should be below the corresponding Figure/Table in the Appendix. Please include a Table of Contents in the beginning of the Appendix. For detailed instructions regarding expanded view please refer to our Author Guidelines: .
- Short and simple tables can be included in the Appendix as Appendix Tables.
- Supplemental files 1 and 2 should be provided as EV Datasets (either as .xls files or .zip folders). Please provide one file per EV Dataset. Please include the description of each EV Dataset in the dataset file itself, ie. in a separate tab for .xls files or as a README.txt file in .zip folders.
- Please provide a "standfirst text" summarizing the study in one or two sentences (approximately 250 characters), three to four "bullet points" highlighting the main findings and a "synopsis image" (exactly 550px width and max 400px height, jpeg format) to highlight the paper on our homepage.
- All Materials and Methods need to be described in the main text. We would ask you to use 'Structured Methods', our new Materials and Methods format, which is mandatory for Methods and Articles with a strong methodological focus. According to this format, the Materials and Methods section should include a Reagents and Tools Table (listing key reagents, experimental models, software and relevant equipment and including their sources and relevant identifiers) followed by a Methods and Protocols section in which we encourage the authors to describe their methods using a step-by-step protocol format with bullet points, to facilitate the adoption of the methodologies across labs. More information on how to adhere to this format as well as downloadable templates (.doc or .xls) for the Reagents and Tools Table can be found in our author guidelines: . An example of a Method paper with Structured Methods can be found here: .
- Please include a "Disclosure and Competing Interests Statement" in the main text.
- Please include a Data availability section describing how the data, code etc. have been made available. This section needs to

be formatted according to the example below:

The datasets and computer code produced in this study are available in the following databases:

- Chip-Seq data: Gene Expression Omnibus GSE46748 (<https://www.ncbi.nlm.nih.gov/geo/query/acc.cgi?acc=GSE46748>)
- Modeling computer scripts: GitHub (<https://github.com/SysBioChalmers/GECKO/releases/tag/v1.0>)
- [data type]: [full name of the resource] [accession number/identifier] ([doi or URL or identifiers.org/DATABASE:ACCESSION])

- The References should be formatted according to the Molecular Systems Biology reference style (i.e. ordered alphabetically and listing the first 10 authors followed by et al).

- For data quantification: please specify the name of the statistical test used to generate error bars and P values, the number (n) of independent experiments (specify technical or biological replicates) underlying each data point and the test used to calculate p-values in each figure legend. The figure legends should contain a basic description of n, P and the test applied. Graphs must include a description of the bars and the error bars (s.d., s.e.m.).

- When you resubmit your manuscript, please download our CHECKLIST (<https://bit.ly/EMBOPressAuthorChecklist>) and include the completed form in your submission.

Please note that the Author Checklist will be published alongside the paper as part of the transparent process (<https://www.embopress.org/page/journal/17444292/authorguide#transparentprocess>).

If you feel you can satisfactorily deal with these points and those listed by the referees, you may wish to submit a revised version of your manuscript. Please attach a covering letter giving details of the way in which you have handled each of the points raised by the referees. A revised manuscript will be once again subject to review and you probably understand that we can give you no guarantee at this stage that the eventual outcome will be favorable.

Yours sincerely,

Maria Polychronidou, PhD
Senior Editor
Molecular Systems Biology

We realize that it is difficult to revise to a specific deadline. In the interest of protecting the conceptual advance provided by the work, we recommend a revision within 3 months (14th Jan 2024). Please discuss the revision progress ahead of this time with the editor if you require more time to complete the revisions. Use the link below to submit your revision:

IMPORTANT: When you send your revision, we will require the following items:

1. the manuscript text in LaTeX, RTF or MS Word format
2. a letter with a detailed description of the changes made in response to the referees. Please specify clearly the exact places in the text (pages and paragraphs) where each change has been made in response to each specific comment given
3. three to four 'bullet points' highlighting the main findings of your study
4. a short 'blurb' text summarizing in two sentences the study (max. 250 characters)
5. a 'thumbnail image' (550px width and max 400px height, Illustrator, PowerPoint or jpeg format), which can be used as 'visual title' for the synopsis section of your paper.
6. Please include an author contributions statement after the Acknowledgements section (see <https://www.embopress.org/page/journal/17444292/authorguide>)
7. Please complete the CHECKLIST available at (<https://bit.ly/EMBOPressAuthorChecklist>).
Please note that the Author Checklist will be published alongside the paper as part of the transparent process (<https://www.embopress.org/page/journal/17444292/authorguide#transparentprocess>).

See also figure legend guidelines: <https://www.embopress.org/page/journal/17444292/authorguide#figureformat>

9. Please note that corresponding authors are required to supply an ORCID ID for their name upon submission of a revised manuscript (EMBO Press signed a joint statement to encourage ORCID adoption).

(<https://www.embopress.org/page/journal/17444292/authorguide#editorialprocess>)

Currently, our records indicate that the ORCID for your account is 0000-0003-4457-1166.

Link Not Available

10. At EMBO Press we ask authors to provide source data for the main manuscript figures. Our source data coordinator will

contact you to discuss which figure panels we would need source data for and will also provide you with helpful tips on how to upload and organize the files.

The system will prompt you to fill in your funding and payment information. This will allow Wiley to send you a quote for the article processing charge (APC) in case of acceptance. This quote takes into account any reduction or fee waivers that you may be eligible for. Authors do not need to pay any fees before their manuscript is accepted and transferred to the publisher.

EMBO Press participates in many Publish and Read agreements that allow authors to publish Open Access with reduced/no publication charges. Check your eligibility: <https://authorservices.wiley.com/author-resources/Journal-Authors/open-access/affiliation-policies-payments/index.html>

*** PLEASE NOTE *** As part of the EMBO Press transparent editorial process initiative (see our Editorial at <https://dx.doi.org/10.1038/msb.2010.72>), Molecular Systems Biology publishes online a Review Process File with each accepted manuscripts. This file will be published in conjunction with your paper and will include the anonymous referee reports, your point-by-point response and all pertinent correspondence relating to the manuscript. If you do NOT want this File to be published, please inform the editorial office at msb@embo.org within 14 days upon receipt of the present letter.

Reviewer #1:

The authors have developed a new AI / machine learning platform to identify sites in pre-mRNAs that can be modulated using splicing switching oligos (SSOs). The authors first used their SpliceCore platform to mine Triple negative breast cancer (TNBC) RNAseq data. They identified hundreds of cassette exons altered in TNBC and confirmed in TNBC cell models. Winnowing of the exons led to a few candidate exons with the authors focusing on exon 13 of NEDD4L due to the correlation of increased survival with inclusion of exon 13. The XGboost model was used to identify binding sites for potential therapeutic SSOs and assess the splicing factor networks (via eCLIP) that are predicted for regulating exon 13 of NEDD4L. Impressively, the authors found one SSO that improved exon 13 inclusion by 67% in cell models. This was accomplished with screening 7 oligos generated from their AI/ML approach.

Strengths of the study include identifying novel targets for TNBC and validation of splicing modulation of NEDD4L using a SSO identified through machine learning (ML) approach. Could dramatically shorten the time to develop oligos to increase exon inclusion (more difficult than exon exclusion). ML approach was tested with known SSOs and they were predicted to work as shown in the literature. Has potential broad applications for many diseases where splice switching strategies could be employed. Overall the manuscript is strong and should be considered for publication.

Two concerns to address: 1) the authors should discuss how their approach could be impacted by RNA structure, which has been shown to play a role in splicing regulation and could impact SSO activity and 2) the authors should strongly consider taking 7 oligos from a different intronic region near exon 13 of NEDD4L and compare the activity of these oligos to those identified through their AI/ML approach.

A minor point for the authors to address:

On page 5 the manuscript states that "XGboost-up showed more borderline, yet clear predictive power with an AUC of 0.67". Predictive power between 0.7 to 0.8 is typically considered acceptable (see reference below), recommend re-wording to "showed borderline predictive power..."

AUC reference

Mandrekar PhD, Jayawant N. "Receiver Operating Characteristic Curve in Diagnostic Test Assessment." *Journal of Thoracic Oncology*, Elsevier, 20 Nov. 2015

Overall, the manuscript was very well written and the authors have developed a powerful tool that will be useful for shortening the time to develop effective SSOs for a range of diseases.

Reviewer #2:

This manuscript describes the development of a machine learning (ML) classifier for the identification of actionable splice-switching oligos (SSOs) for experimental and potentially therapeutic applications. The authors have trained ML models using massively parallel splicing reporter assays as a proxy to infer the impact of SSOs on pre-mRNA splicing and by combining regulatory information derived from splicing factor binding motifs (based on published in vitro experiments) as well as splicing factor interactions. Reassuringly, the developed ML model performed well using independently derived reporter assay experiments, particularly for intronic SSO targeting. Feature selection methods reveal that the distance from splice site is the most predictive feature for the intronic models. The authors apply their model to predict efficient SSOs of nine previously experimentally interrogated alternative splicing events. Although the data are not presented in a very clear format, the authors claim that their model selects previously validated SSOs and predicts the splicing regulators that underlie their effect. The authors then applied a proprietary RNA-Seq analytical pipeline to TCGA data in order to identify alternative splicing events associated with triple negative breast cancer and further selected a more confident set of events by generating RNA-seq data from corresponding cell lines. The authors select one of the strongest candidate exons, the alternatively spliced exon-13 in NEDD4L, for detailed characterization. Their analysis predicts that SRSF7 and HNRNPL can regulate this event (something that is not confirmed experimentally). Finally, the authors identify an efficient SSO (0205) that promote NEDD4L exon-13 inclusion. The last part of the paper focuses on interrogating the effect of SSO-0205 in TGF β -dependent proliferative and migratory responses.

This is a potentially interesting study, and the developed ML model could be proven useful for the community to identify SSOs and key splicing regulators for phenotypically relevant splicing events. However, it appears that this tool will remain proprietary limiting its widespread use for the community. A weakness of the manuscript in its current form is the lack of sufficient experimental validation to the predictions of the ML model. The NEDD4L study is interesting but the quality of the data is not utterly convincing for the conclusions that the authors make. Specific major and minor comments follow:

Major comments

1. The manuscript lacks sufficient validations at multiple instances. Most importantly, the authors should confirm the impact of SRSF7 and HNRNPL on NEDD4L exon-13 splicing using siRNA experiments (and ideally rescue any effect by adding back siRNA resistant cDNA of the targeted genes). Furthermore, the authors should validate the splicing factor predictions of the nine events shown in Figure 2.
2. Related to the previous comment, it is unclear to the reader to what extent the ML model effectively predicts the most efficient SSO for the nine interrogated splicing events presented in Figure 2. Given that the tested exons have been previously targeted by dozens SSOs as the authors state, the authors should aim to present a figure summarizing the efficiency of their approach for identifying the most efficient SSOs (e.g. correlation plot or ROC curve). Furthermore, the predictive data generated by the ML model for these nine events should be provided in a supplementary table.
3. The SpliceCore derived splicing events of interest as shown in Figures 3A and B should be included in a Supplementary Table. If published, the manuscript should contain all the identified differentially spliced events between the different TCGA breast cancer datasets and related cell lines. The table should also include the type of alternative splicing event, genomic coordinates, Δ PSI values and other relevant information.
4. The authors claim that SSO-0205 treatment following TGF β stimulation results in subcellular localization changes of several proteins. This is difficult to appreciate by the provided western blots and corresponding quantifications and such conclusions should be further validated by immunofluorescence analyses.
5. The Supplementary Figure 7A lacks appropriate controls. The authors should include minus MG132 treatment side-by-side to the MG132-treated cells and also should include a negative control IP to assess the specificity of their immunoprecipitation assays.

Minor comments

1. The authors should mention in the introduction or discussion other related ML approaches for SSO selection (e.g. PMID: 34104972; PMID: 37513994).
2. The description of the ML model development, selection and code is not sufficiently described in the manuscript and more information should be included in the methods section. For example, what are the parameters and software/packages used to implement the XGboost models (XGboost-down, XGboost-up and XGboost-exon). Also, were the same feature sets used for the other five explored models?
3. The authors should explain how they define SF interactions applied in their ML model and why they are not using experimentally validated PPIs from BIOGRID and/or STRING for their analyses.
4. The SHAP/OOB analysis should also be performed in the XGboost-exon model.

5. The authors should explain what the cluster suffix represents (i.e.: *_c1, *_c2 and *_nc) in Supplementary File 1.
6. In the methods section, mention and describe in more details the applied performance metrics (i.e. sensitivity, specificity, and AUC) for the convenience of non-expert readers.
7. The explanation of why SSO-0205 only works in MDA-MB-231 and not in MCF10A cells is hard to understand given that both hnRNPL and SRSF7 are expressed in both cell lines (Figure 7B). These data suggest that either there is another negative regulator of this exon expressed in MCF10A or else that a potential positive regulator of this exon is only expressed in MDA-MB-231 cells.
8. Page 9: "... a cell cycle evaluation indicated that SSO-0205 treatment significantly increased the number of cells in G1 and decreased the number of cells in G2 at the 24-hour time point (Figure 7D)".
A statistical test should be performed and mentioned along with a p-value in Figure 7D legend for both MDA-MB-231 and MCF10A cell-cycle analysis data.
9. As mentioned in major comment-4 the data describing the impact of NEDDL4 SSO-0205 on the localization of TGF β pathway members is extremely difficult to follow. The authors should break down their conclusions to sub-panels that need to be referred in the text rather than just citing "Supplementary Figure 6" (which contains ~15 panels!), at the very end of a long paragraph.
10. Page 9; Line 9: "MMG132" should be "MG132".
11. Supplementary Figure 4 legend (three lines from the end): "rAMTs" should be "rMATs". Also, rMATs should be rMATs in the actual figure.
12. Supplementary Figure 7A: "SSO02-05" should be "SSO-0205".
13. Root mean square deviation (presumably) should be spelt out in the legend of Figure 4.
14. There is inconsistent use of XGboost-down/XGboost-downstream and XGboost-up/ XGboost-upstream throughout the manuscript.

Dear Dr. Polychronidou,

I hope this message finds you well! I am writing to express my sincere appreciation for your consideration of our manuscript submitted to Molecular Systems Biology. The reviewer's insightful comments and valuable feedback have been instrumental in refining our work. I am pleased to inform you that we have diligently addressed all reviewer comments (detailed below), incorporating necessary revisions to enhance the overall quality of the manuscript. We believe these changes have significantly strengthened the clarity and impact of our research. I look forward to the possibility of our work being published in Molecular Systems Biology.

Reviewer #1 major comments (reviewer comments in italics):

Comment 1a: Two concerns to address: 1) the authors should discuss how their approach could be impacted by RNA structure, which has been shown to play a role in splicing regulation and could impact SSO activity and 2) the authors should strongly consider taking 7 oligos from a different intronic region near exon 13 of NEDD4L and compare the activity of these oligos to those identified through their AI/ML approach.

Author response 1a: We thank the reviewer for their comment. Indeed, RNA structure plays a critical role in determining the success of SSOs in modulating splicing. During the initial stages of development of the model in this manuscript, we attempted to incorporate RNA secondary structure as a predictive feature. However, our analysis did not reveal any significant contributions from stem-loop structures. This observation aligns with insights from Kate Cook, Tim Hughes, and Quaid Morris, as highlighted in their 2015 review (Cook et al., 2015 Briefings in Functional Genomics) wherein they point out "For certain RBPs, RNA secondary structure is a key component of RNA binding, but widespread prediction and measurement of RNA secondary structure remains difficult especially due to the potential impact of other RBPs on mRNA structure". Perhaps this observation underscores the challenge of accurately capturing RNA secondary structure's contribution into our model. Furthermore, the same review notes that "Despite [methods for RBP binding prediction] ignoring secondary structure, these methods are often successful, presumably because many RBPs fundamentally bind short ssRNA sequences (5–10 nt)", suggesting that secondary structure may not always be a decisive factor. Additionally, previous research (Hiller et al., 2007 PLoS Genetics) has demonstrated that the absence of secondary structure (i.e., RNA linearity) can serve as a strong predictor for RBP binding. Finally, it is important to note that despite optimism expressed by many experts in the field, we rigorously assessed the impact of secondary structure on predictability and found consistent results, wherein this factor did not significantly influence outcomes. We have added text and a reference in the second paragraph of the discussion (page 12) considering this point.

In response to the second part of this comment, we wholeheartedly concur with the author's perspective. Consequently, we embarked on a test involving seven control SSOs sourced from nearby intronic regions. This successfully demonstrated that SSOs predicted to be inactive in the downstream intron indeed had no discernible effect on NEDD4Le13 splicing across all tested cell lines (Appendix Figure S5).

Reviewer #1 minor comments (reviewer comments in italics):

Comment 1b: A minor point for the authors to address: On page 5 the manuscript states that "XGboost-up showed more borderline, yet clear predictive power with an AUC of 0.67". Predictive power between 0.7 to 0.8 is typically considered acceptable (see reference below), recommend re-wording to "showed borderline predictive power... AUC reference: Mandrekar PhD, Jayawant N. "Receiver Operating Characteristic Curve in Diagnostic Test Assessment." Journal of Thoracic Oncology, Elsevier, 20 Nov. 2015

Author response 1b: We thank the reviewer for their comment and have revised the manuscript as suggested (page 6).

Reviewer #2 major comments (reviewer comments in italics):

Comment 2a: This is a potentially interesting study, and the developed ML model could be proven useful for the community to identify SSOs and key splicing regulators for phenotypically relevant splicing events. However, it appears that this tool will remain proprietary limiting its widespread use for the community. A

weakness of the manuscript in its current form is the lack of sufficient experimental validation to the predictions of the ML model. The NEDD4L study is interesting but the quality of the data is not utterly convincing for the conclusions that the authors make.

Author response 2a: We thank the reviewer for their comments. In our efforts to enhance the accessibility and reproducibility of our methodologies, we have taken several steps. Firstly, we've made the training data for our models available as a supplemental table (Dataset EV2). Additionally, we have introduced a new section within the Methods (pages 14-15), titled "XGboost Model Development," which provides a comprehensive breakdown of our methodology. This section, spanning an entire page, is divided into four sub-parts: SFs used for feature engineering, Differential SF binding estimation, Splicing factor clusters (SFCs) compilation, and Machine Learning software and models. We believe that this detailed information, along with the insights into the SpliceCore platform presented in the Appendix section, will greatly assist readers in implementing our approach. It's worth noting that our manuscript does not introduce a new software package or bioinformatics tool. Rather, our aim was to leverage open-source ML packages like Scikit Learn and Scipy in Python and apply them to RBP data. Throughout the manuscript, we emphasized our goal of demonstrating that an AI/ML approach to RNA therapeutics can yield novel and actionable insights. We believe we've achieved this objective both retrospectively, through successful testing of previously discovered SSOs, and prospectively, by uncovering a new disease-related splicing event. Additionally, we've showcased how AI/ML assistance in designing an SSO led to the development of an effective treatment against cancer cells. We believe this approach aligns with numerous strategies for analyzing genomic data and Molecular Systems Biology policies to provide sufficient detail to allow reproduction of the underlying algorithms by others. Specifically, Molecular Systems Biology policies state "This may be achieved by providing the appropriate narrative or mathematical description, pseudocode and, possibly but not obligatorily, the source code." Finally, we would like to add that it is important to us that others will find our approach useful and that we are happy to provide assistance and communication with anyone looking to use or emulate our approach.

Additionally, we have done further experimental studies as suggested by the reviewer, including siRNA treatment to confirm the effect of the predicted SFs on NEDD4Le13 (Figure EV3), testing SSOs not predicted to effect NEDD4Le13 splicing (Appendix Figure S5), additional controls on the IP and protease inhibitor treatments (Appendix Figure S6), and statistical analysis on the cell cycle study data (Figure 7D). Additional details regarding the experimental validation of our model are provided in the responses below.

Comment 2b: The manuscript lacks sufficient validations at multiple instances. Most importantly, the authors should confirm the impact of SRSF7 and HNRNPL on NEDD4L exon-13 splicing using siRNA experiments (and ideally rescue any effect by adding back siRNA resistant cDNA of the targeted genes). Furthermore, the authors should validate the splicing factor predictions of the nine events shown in Figure 2.

Author response 2b: We thank the reviewer for suggesting siRNA experiments and have conducted siRNA knockdown to confirm the impact of SRSF7 and HNRNPL on NEDD4L (Figure EV3). Section 7 of the results "AI/ML identified an optimal SSO to promote NEDD4Le13 inclusion and explained the underlying AS regulatory network" summarizes these additions (page 9). Regarding the nine events shown in Figure 2, the splicing regulatory effect of these SSOs has already been validated in the cited publications. We strongly believe the knowledge to be gained from repeating these studies would not justify the time and resources required to develop 9 new cell model systems to test all events and RBPs appropriately. The data provided supports the use of RBP predictions to achieve high predictive power across several events (see comment 2c, and Figure EV2).

Comment 2c: Related to the previous comment, it is unclear to the reader to what extent the ML model effectively predicts the most efficient SSO for the nine interrogated splicing events presented in Figure 2. Given that the tested exons have been previously targeted by dozens SSOs as the authors state, the authors should aim to present a figure summarizing the efficiency of their approach for identifying the most efficient SSOs (e.g. correlation plot or ROC curve). Furthermore, the predictive data generated by the ML model for these nine events should be provided in a supplementary table.

Author response 2c: We thank the reviewer for their comment and the revised manuscript now includes a figure summarizing the efficiency of finding the most effective SSO for benchmarking events along with a figure describing the predictive data (including the alternative splicing effect probability and the most predictive splicing factors at each position in the alternative splicing event) generated for these 9 events (Figure EV2 and Dataset EV3). Section 4 of the results “XGboost model retrospectively identifies SSO known to modulate AS events” summarizes these additions (page 7).

Comment 2d: The SpliceCore derived splicing events of interest as shown in Figures 3A and B should be included in a Supplementary Table. If published, the manuscript should contain all the identified differentially spliced events between the different TCGA breast cancer datasets and related cell lines. The table should also include the type of alternative splicing event, genomic co-ordinates, Δ PSI values and other relevant information.

Author response 2d: The revised manuscript now includes a table of SpliceCore- identified differentially spliced events between the different TCGA breast cancer datasets and related cell lines that contain data on the splicing events, coordinates, PSI values, and the reproducibility and consistency of each event across samples (Dataset EV4).

Comment 2e: The authors claim that SSO-0205 treatment following TGF β stimulation results in subcellular localization changes of several proteins. This is difficult to appreciate by the provided western blots and corresponding quantifications and such conclusions should be further validated by immunofluorescence analyses.

Author response 2e: Though we appreciate the reviewer’s point and we understand that other publications show immunofluorescence (IF) images for flagged or tagged SMAD proteins to visualize subcellular localization, we are confident that the data provided is sufficient to validate our subcellular localization claims under naïve conditions (not tagged). Western blots give a better quantitative profile of steady-state protein levels in different compartments than IF and it is difficult to assess steady-state levels of rapidly shuttling proteins such as SMADs with IF. This is a common experiment in TGF β field to look at SMAD accumulation (Liu et al., 2016 Scientific Reports, Figures 1 and 2; Frick et al., 2017 PNAS, Figure S2) In addition, the phosphospecific antibodies used in the assay work better on Western blot compared to immunofluorescence. Therefore, IF it is not an appropriate assay to measure the subcellular distribution of the proteins under our experimental conditions. We believe that we have provided sufficient controls to assess the purity of the fractionated cellular compartments to demonstrate that the change in distribution is biologically significant.

Comment 2f: The Supplementary Figure 7A lacks appropriate controls. The authors should include minus MG132 treatment side-by-side to the MG132-treated cells and also should include a negative control IP to assess the specificity of their immunoprecipitation assays.

Author response 2f: We thank the reviewer for their comment and have included the specified negative control experiments which can be found in Appendix Figure S6.

Reviewer #2 minor comments (reviewer comments in italics):

Comment 2g: The authors should mention in the introduction or discussion other related ML approaches for SSO selection (e.g. PMID: 34104972; PMID: 37513994).

Author response 2g: The revised manuscript now mentions and cites other ML approaches in the second paragraph of the discussion (page 12).

Comment 2h: The description of the ML model development, selection and code is not sufficiently described in the manuscript and more information should be included in the methods section. For example, what are the parameters and software/packages used to implement the XGboost models (XGboost-down, XGboost-up and XGboost-exon). Also, were the same feature sets used for the other five explored models?

Author response 2h: To make our methodologies accessible and reproducible, we have made the training data for our models available as a supplemental table (Dataset EV2) and have described our methodologies

thoroughly in the manuscript and appendix. Furthermore, the manuscript includes a detailed description of XGboost implementation. The same feature sets were used for all models tested.

Comment 2i: The authors should explain how they define SF interactions applied in their ML model and why they are not using experimentally validated PPIs from BIOGRID and/or STRING for their analyses.

Author response 2i: The SF interactions are defined as they were in a previous manuscript published by the senior author, Dr. Akerman. We elaborate on this in the first results section (page 4) of the manuscript (“An AI/ML model to predict SSO binding sites for AS modulation”). In brief, the SF interactions used in this manuscript take both interactome and gene expression data into account and are validated experimentally, as described in Akerman 2015. By using the probabilistic approach to determine the protein-protein interactions, it is possible to incorporate the dynamic nature of the spliceosome into the SF interaction data and therefore into our features.

Comment 2j: The SHAP/OOB analysis should also be performed in the XGboost-exon model.

Author response 2j: Dataset EV1 highlights the contribution of the various SFCs to the XGboost-exon model. However, because the XGboost-exon model showed modest predictive power it was not used to evaluate any splicing event for functional SSOs, and therefore applying the SHAP/OOB analysis would not yield any useful results. This strategy was employed specifically to find the features that are most predictive for a specific SSO result.

*Comment 2k: The authors should explain what the cluster suffix represents (i.e.: *_c1, *_c2 and *_nc) in Supplementary File 1.*

Author response 2k: Cluster suffixes are now explained in the “key” tab of Dataset EV1.

Comment 2l: In the methods section, mention and describe in more details the applied performance metrics (i.e. sensitivity, specificity, and AUC) for the convenience of non-expert readers.

Author response 2l: The revised manuscript explains these metrics in the “Machine learning software and models” section in the Materials and Methods (page 15).

Comment 2m: The explanation of why SSO-0205 only works in MDA-MB-231 and not in MCF10A cells is hard to understand given that both hnRNPL and SRSF7 are expressed in both cell lines (Figure 7B). These data suggest that either there is another negative regulator of this exon expressed in MCF10A or else that a potential positive regulator of this exon is only expressed in MDA-MB-231 cells.

Author response 2m: Many RBPs are multi-functional and behave differently in healthy versus tumorigenic conditions. We have included these points in the “SSOs targeted to NEDD4Le13 promoted exon inclusion and affected the TGF β pathway” section of the results (page 10) along with relevant citations.

Comment 2n: Page 9: "... a cell cycle evaluation indicated that SSO-0205 treatment significantly increased the number of cells in G1 and decreased the number of cells in G2 at the 24-hour time point (Figure 7D)".

A statistical test should be performed and mentioned along with a p-value in Figure 7D legend for both MDA-MB-231 and MCF10A cell-cycle analysis data.

Author response 2n: The statistical method used (One-way ANOVA) and p-values for both cell lines were added to the legend of Figure 7.

Comment 2o: As mentioned in major comment-4 the data describing the impact of NEDDL4 SSO-0205 on the localization of TGF β pathway members is extremely difficult to follow. The authors should break down their conclusions to sub-panels that need to be referred in the text rather than just citing "Supplementary Figure 6" (which contains ~15 panels!), at the very end of a long paragraph.

Author response 2o: We thank the reviewer for their comment. We have added detailed figure references to the text in the results sections to connect the different statements with the corresponding Western blots in the

figure. We have also added squares with dashed lines to highlight different time points in the Western blot images to improve the interpretation of the results.

Comment 2p: Page 9; Line 9: "MMG132" should be "MG132".

Author response 2p: We thank the reviewer for their attention to detail and have revised the text accordingly.

Comment 2q: Supplementary Figure 4 legend (three lines from the end): "rAMTs" should be "rMATS". Also, rMATs should be rMATS in the actual figure.

Author response 2q: We thank the reviewer for their attention to detail and have revised the figure accordingly.

Comment 2r: Supplementary Figure 7A: "SSO02-05" should be "SSO-0205".

Author response 2r: We thank the reviewer for their attention to detail and have revised the figure accordingly.

Comment 2s: Root mean square deviation (presumably) should be spelt out in the legend of Figure 4.

Author response 2s: We thank the reviewer for their attention to detail and have revised the legend accordingly.

Comment 2t: There is inconsistent use of XGboost-down/XGboost-downstream and XGboost-up/ XGboost-upstream throughout the manuscript.

Author response 2t: We thank the reviewer for their attention to detail and have revised the manuscript for consistency.

In summary, we believe our revisions have addressed all of the reviewer comments and that this manuscript will appeal to the readership of Molecular Systems Biology because it presents experimental proof for the use of AI/ML applied to RNA therapeutics and thus aligns with the computational biology core topic of this journal. The novel computational approaches presented in this manuscript add explainability to the insights gained from the large-scale datasets analyzed, further aligning with the journal's scope. Presenting concrete evidence of the benefit that AI/ML can have on RNA therapeutic development is crucial to increasing the scientific community's trust and reliance on predictive approaches. To make our methodologies accessible and reproducible, we have made the training data for our models available as an EV dataset and have described our methodologies thoroughly in the manuscript and appendix.

We confirm that this manuscript has not been published elsewhere and is not under consideration by another journal, although a version of the manuscript is available on BioRxiv (Fronk AD, et. al., 2022). All authors have approved the manuscript and agree with its submission to Molecular Systems Biology.

Thank you for your consideration. We welcome the opportunity to publish in your journal and contribute to the ongoing scholarship.

Sincerely,

Martin Akerman, Ph.D.
CTO, Envisagenics, Inc.

15th Feb 2024

Manuscript Number: MSB-2023-12002R

Title: Development and validation of AI/ML derived splice-switching oligonucleotides

Dear Dr. Akerman,

Thank you for submitting your revised manuscript. I have now gone through the revised manuscript and your point by point responses and discussed them with the team. One of the issues raised during the evaluation of the initial version of the study, referred to the restrictions in the availability of the model, which will limit its use by the community. This issue was raised both by reviewer #2 and editorially. As I will explain below, we do not think that this issue has been addressed satisfactorily.

We noted that in the revised version of the manuscript the final trained ML model(s) have not been made available. In your point-by-point response you mention that you have i) made the training data for the models available in Dataset EV2 and ii) included further details regarding the model development in the Methods and Appendix. However, this would mean that readers who would like to use the ML model would need to reconstruct it themselves. As per our author guidelines, we require that data, models, software, code etc. that are integral to the study should be made available without restrictions and in a form that allows implementation by others and integration in other workflows. This requirement and any restrictions related to it, as mentioned in our guidelines, is taken into account in our editorial decision in combination with the advance presented by the study, the potential utility of the presented tool etc. and may even prevent the publication of the study.

In this case we feel that the presented platform (ML model) is central and integral to the study. While the source code may not be necessarily relevant, we think that the trained model should be made available for reuse and integration in other projects. We do not think that having to reconstruct the model from the instructions/information/training data provided in the paper seems sufficient. As such, before we can send the study back to the reviewers to evaluate the performed revisions, we would ask you to make the model available. The information on where/how the model has been made available should be included in the Data Availability section.

You can submit your revised manuscript here:

Kind regards,

Maria

Maria Polychronidou, PhD
Senior Editor
Molecular Systems Biology

If you do choose to resubmit, please click on the link below to submit the revision online before 14th Mar 2024.

***** PLEASE NOTE ***** As part of the EMBO Press transparent editorial process initiative (see our Editorial at <https://dx.doi.org/10.1038/msb.2010.72> , Molecular Systems Biology will publish online a Review Process File to accompany accepted manuscripts. When preparing your letter of response, please be aware that in the event of acceptance, your cover letter/point-by-point document will be included as part of this File, which will be available to the scientific community. More information about this initiative is available in our Instructions to Authors. If you have any questions about this initiative, please contact the editorial office (msb@embo.org).

The authors addressed the requested editorial issues.

22nd Mar 2024

Manuscript Number: MSB-2023-12002RR

Title: Development and validation of AI/ML derived splice-switching oligonucleotides

Dear Dr. Akerman,

Thank you for sending us your revised manuscript. We have now heard back from the two reviewers who were asked to evaluate your revised study. As you will see below, the reviewers are satisfied with the performed revisions and support publication. As such, I am glad to inform you that we can soon accept your study for publication, pending some minor revisions related to the editorial issues listed below.

- In line with the comment of reviewer #2, we would ask you to provide the 4 EV Datasets that are referenced in the text (Dataset EV1-EV4). Please provide one file per EV Dataset. Please include the description of each EV Dataset in the dataset file itself, i.e. in a separate tab for .xls files or as a README.txt file in .zip folders for .csv files.
- Our data editors have noticed some unclear or missing information in the figure legends.
 - Please indicate the statistical test used for data analysis in the legends of figures 5b.
 - Please note that for the figures 6b, 7g p-values and statistical tests are indicated in the legends. However, comparison for the same, ""#/***/"" has not been represented in the figures. Please rectify this in the figures or legends as applicable.
 - The box plots need to be defined in terms of minima, maxima, centre, bounds of box and whiskers, and percentile in the legends of figures 4e-f; 5c-d.
 - Please define the error bars in the legends of figures EV 3c; EV 4e."
- The primer table needs a number e.g. Table 1.
- The References should be formatted according to the Molecular Systems Biology reference style (i.e., ordered alphabetically and listing the first 10 authors followed by et al).
- Please remove the 'Authors Contributions' from the manuscript. The 'Author Contributions' section is replaced by the CRediT contributor roles taxonomy to specify the contributions of each author in the journal submission system. Please use the free text box in the 'author information' section of the online submission system to provide more detailed descriptions if needed (e.g., 'X provided intracellular Ca⁺⁺ measurements in fig Y').
- Appendix table 1 should be renamed to Appendix Table S1 in the Table of Contents and the legend below the table
- The Source Data files need to be reorganized. Please provide to one file (or zip folder) per figure. All Source Data for EV and/or Appendix figures should be provided in a single zip folder.
- The synopsis image is rather large and detailed and not all labels display well at the final size required. Please resupply the image as a jpg or png at the required final size (it needs to be exactly 550 px wide, and the height ideally < 500 px), ensuring that all labels are legible. We would recommend reducing the amount of text on the figure as much as possible.
- Our data integrity analyst noted that the same immunoblot image for cytoplasmic Actin is used twice in Figure EV4A. The source data is also identical. Please clarify the image re-use. It also needs to be indicated in the figure legend.

Please resubmit your revised manuscript online, with a covering letter listing amendments and responses to each point raised by the referees. Please resubmit the paper ****within one month**** and ideally as soon as possible. If we do not receive the revised manuscript within this time period, the file might be closed and any subsequent resubmission would be treated as a new manuscript. Please use the Manuscript Number (above) in all correspondence.

Click on the link below to submit your revised paper.

Yours sincerely,

Maria Polychronidou, PhD
Senior Editor

If you do choose to resubmit, please click on the link below to submit the revision online before 21st Apr 2024.

IMPORTANT:

Please note that corresponding authors are required to supply an ORCID ID for their name upon submission of a revised manuscript (EMBO Press signed a joint statement to encourage ORCID adoption).
(<https://www.embopress.org/page/journal/17444292/authorguide#editorialprocess>)
Currently, our records indicate that the ORCID for your account is 0000-0003-4457-1166.

Please click the link below to modify this ORCID:
Link Not Available

*** PLEASE NOTE *** As part of the EMBO Press transparent editorial process initiative (see our Editorial at <https://dx.doi.org/10.1038/msb.2010.72> , Molecular Systems Biology will publish online a Review Process File to accompany accepted manuscripts. When preparing your letter of response, please be aware that in the event of acceptance, your cover letter/point-by-point document will be included as part of this File, which will be available to the scientific community. More information about this initiative is available in our Instructions to Authors. If you have any questions about this initiative, please contact the editorial office (msb@embo.org).

Reviewer #1:

The authors have addressed my comments.

Reviewer #2:

The authors have adequately addressed the reviewers' comments, thereby rendering the manuscript suitable for publication. However, I have a couple of minor concerns: this reviewer was unable to locate the EV datasets referenced by the authors in the response letter and revised manuscript. Also the main and EV figures are not labelled make it a little challenging to navigate through the manuscript. I trust that the handling editor will ensure that all necessary components are in place as described by the authors before proceeding with publication.

Dear Dr. Polychronidou,

I hope this message finds you well! I am writing to express my sincere appreciation for supporting the publication of our manuscript in Molecular Systems Biology. We are grateful for the reviewer's insightful comments which in addressing we believe have led to a more impactful manuscript for the scientific community. We have addressed all the editorial revisions suggested through your latest communication and provide a detailed list of the manuscript amendments below.

Editorial issue #1 (issue in italics):

Issue 1: In line with the comment of reviewer #2, we would ask you to provide the 4 EV Datasets that are referenced in the text (Dataset EV1-EV4). Please provide one file per EV Dataset. Please include the description of each EV Dataset in the dataset file itself, i.e. in a separate tab for .xls files or as a README.txt file in .zip folders for .csv files.

Author response 1: We have included the requested files which are named MSB-2023-12002R_DatasetEV*, corresponding to each of the 4 EV datasets.

Editorial issue #2 (issue in italics):

Issue 2: Our data editors have noticed some unclear or missing information in the figure legends.

2a. Please indicate the statistical test used for data analysis in the legends of figures 5b.

Author response 2a: We have updated the legend of figure 5b to indicate the statistical tests used.

*2b. Please note that for the figures 6b, 7g p-values and statistical tests are indicated in the legends. However, comparison for the same, ""#/***/*"" has not been represented in the figures. Please rectify this in the figures or legends as applicable.*

Author response 2b: We have updated figures 6 and 7 so that the p-values in the legend and symbols in the figures match.

2c. The box plots need to be defined in terms of minima, maxima, centre, bounds of box and whiskers, and percentile in the legends of figures 4e-f; 5c-d.

Author response 2c: We have updated figures 4e-f and 5c-d to define the minima, maxima, center, and bounds of the box plots.

2d. Please define the error bars in the legends of figures EV 3c; EV 4e.

Author response 2d: We have updated figures EV 3c and EV 4e to define the error bars.

Editorial issue #3 (issue in italics):

Issue 3: The primer table needs a number e.g. Table 1.

Author response 3: The primer table is now numbered Table 1.

Editorial issue #4 (issue in italics):

Issue 4: The References should be formatted according to the Molecular Systems Biology reference style (i.e., ordered alphabetically and listing the first 10 authors followed by et al).

Author response 4: We have updated the references to match Molecular Systems Biology style.

Editorial issue #5 (issue in italics):

Issue 5: Please remove the 'Authors Contributions' from the manuscript. The 'Author Contributions' section is replaced by the CRediT contributor roles taxonomy to specify the contributions of each author in the journal submission system. Please use the free text box in the 'author information' section of the online submission system to provide more detailed descriptions if needed (e.g., 'X provided intracellular Ca⁺⁺ measurements in fig Y').

Author response 5: We have removed the author contributions as requested.

Editorial issue #6 (issue in italics):

Issue 6: Appendix table 1 should be renamed to Appendix Table S1 in the Table of Contents and the legend below the table.

Author response 6: We have renamed the appendix table as requested.

Editorial issue #7 (issue in italics):

Issue 7: The Source Data files need to be reorganized. Please provide to one file (or zip folder) per figure. All Source Data for EV and/or Appendix figures should be provided in a single zip folder.

Author response 7: Source data files have been reorganized into a single file for Figure 3C and Figure 4A, a single zip folder for Figure 6, Figure 7 and a single zip folder for all Expanded View figures.

Editorial issue #8 (issue in italics):

Issue 8: The synopsis image is rather large and detailed and not all labels display well at the final size required. Please resupply the image as a jpg or png at the required final size (it needs to be exactly 550 px wide, and the height ideally < 500 px), ensuring that all labels are legible. We would recommend reducing the amount of text on the figure as much as possible.

Author response 8: We have edited and adjusted the synopsis image as requested.

Editorial issue #9 (issue in italics):

Issue 9: Our data integrity analyst noted that the same immunoblot image for cytoplasmic Actin is used twice in Figure EV4A. The source data is also identical. Please clarify the image re-use. It also needs to be indicated in the figure legend.

Author response 9: We have updated the figure legend to indicate that cytoplasmic actin is shared for Cytoplasmic SMAD2/3 and phSMAD2/3, since the membrane was stripped and re-probed.

We again extend our sincerest appreciation for supporting the publication of our manuscript in Molecular Systems Biology.

We confirm that this manuscript has not been published elsewhere and is not under consideration by another journal, although a version of the manuscript is available on BioRxiv (Fronk AD, et. al., 2022). All authors have approved the manuscript and agree with its submission to Molecular Systems Biology.

Sincerely,

Martin Akerman, Ph.D.
CTO, Envisagenics, Inc.

9th Apr 2024

Manuscript number: MSB-2023-12002RRR

Title: Development and validation of AI/ML derived splice-switching oligonucleotides

Dear Dr. Akerman,

Thank you again for sending us your revised manuscript. We are now satisfied with the modifications made and I am pleased to inform you that your paper has been accepted for publication.

Yours sincerely,

Maria Polychronidou, PhD
Senior Editor
Molecular Systems Biology
